
# Uncertainties in eddy covariance air-sea $CO_2$ flux measurements and implications for gas transfer velocity parameterisations

**Yuanxu Dong[1,2], Mingxi Yang[2], Dorothee C. E. Bakker[1], Vassilis Kitidis[2] and Thomas G. Bell[2]**

[1]Centre for Ocean and Atmospheric Sciences, School of Environmental Sciences, University of East Anglia, Norwich, UK

[2]Plymouth Marine Laboratory, Prospect Place, Plymouth, UK

*Correspondence to:* Yuanxu Dong (Yuanxu.Dong@uea.ac.uk)

**Abstract.** Air-sea carbon dioxide ($CO_2$) flux is often indirectly estimated by the bulk method using the air-sea difference in $CO_2$ fugacity ($\Delta fCO_2$) and a parameterisation of the gas transfer velocity ($K$). Direct flux measurements by eddy covariance (EC) provide an independent reference for bulk flux estimates and are often used to study processes that drive $K$. However, inherent uncertainties in EC air-sea $CO_2$ flux measurements from ships have not been well quantified and may confound analyses of $K$. This paper evaluates the uncertainties in EC $CO_2$ fluxes from four cruises. Fluxes were measured with two state-of-the-art closed-path $CO_2$ analysers on two ships. The mean bias in the EC $CO_2$ flux is low but the random error is relatively large over short time scales. The uncertainty (1 standard deviation) in hourly averaged EC air-sea $CO_2$ fluxes (cruise-mean) ranges from 1.4 to 3.2 mmol m$^{-2}$ day$^{-1}$. This corresponds to a relative uncertainty of ~20% during two Arctic cruises that observed large $CO_2$ flux magnitude. The relative uncertainty was greater (~50%) when the $CO_2$ flux magnitude was small during two Atlantic cruises. Random uncertainty in the EC $CO_2$ flux is mostly caused by sampling error. Instrument noise is relatively unimportant. Random uncertainty in EC $CO_2$ fluxes can be reduced by averaging for longer. However, averaging for too long will result in the inclusion of more natural variability. Auto-covariance analysis of $CO_2$ fluxes suggests that the optimal timescale for averaging EC $CO_2$ flux measurements ranges from 1–3 hours, which increases the mean signal-to-noise ratio of the four cruises to higher than 3. Applying an appropriate averaging timescale and suitable $\Delta fCO_2$ threshold (20 µatm) to EC flux data enables an optimal analysis of $K$.



## 1 Introduction

Since the Industrial Revolution, atmospheric $CO_2$ levels have risen steeply due to human activities (Broecker and Peng, 1993). The ocean plays a key role in the global carbon cycle, having taken up roughly one quarter of anthropogenic $CO_2$ emissions over the last decade (Friedlingstein et al., 2020). Accurate estimates of air-sea $CO_2$ flux are vital to forecast climate change and to quantify the effects of ocean $CO_2$ uptake on the marine biosphere.

Air-sea $CO_2$ flux ($F$, e.g. in mmol m$^{-2}$ day$^{-1}$) is typically estimated indirectly by the bulk equation:

$$F = K_{660}(Sc/660)^{-0.5}\, \alpha\left(f\mathrm{CO}_{2\mathrm{w}} - f\mathrm{CO}_{2\mathrm{a}}\right) \tag{1}$$

Where $K_{660}$ (in cm h$^{-1}$) is the gas transfer velocity, usually parameterised as a function of wind speed (e.g. Nightingale et al., 2000), $Sc$ (dimensionless) is the Schmidt number (Wanninkhof, 2014) and $\alpha$ (mol L$^{-1}$ atm$^{-1}$) is the solubility (Weiss, 1974). $f\mathrm{CO}_{2\mathrm{w}}$ and $f\mathrm{CO}_{2\mathrm{a}}$ are the $CO_2$ fugacity (in µatm) at the sea surface and in the overlying atmosphere, respectively, with $f\mathrm{CO}_{2\mathrm{w}} - f\mathrm{CO}_{2\mathrm{a}}$ the air-sea $CO_2$ fugacity difference ($\Delta f\mathrm{CO}_2$). Uncertainties in the $K_{660}$ parameterisation and limited coverage of $f\mathrm{CO}_{2\mathrm{w}}$ measurements result in considerable uncertainties in global bulk flux estimates (Takahashi et al., 2009; Woolf et al., 2019).

Eddy covariance (EC) is the most direct method for measuring the air-sea $CO_2$ flux $F$:

$$F = \rho\,\overline{w'c'} \tag{2}$$

where $\rho$ is the mean mole density of dry air (e.g. in mole m$^{-3}$). The dry $CO_2$ mixing ratio $c$ (in ppm or µmol mol$^{-1}$) is measured by a fast-response gas analyser and the vertical wind velocity $w$ (in m s$^{-1}$) is often measured by a sonic anemometer. The prime denotes the fluctuations from the mean, while the overbar indicates time average. Equation 2 does not rely on $\Delta f\mathrm{CO}_2$ measurements nor empirical parameters and assumptions of the gas properties (Wanninkhof, 2014). EC flux measurements can therefore be considered useful as an independent reference for bulk air-sea $CO_2$ flux estimates. Furthermore, the typical temporal and spatial scales of EC flux measurements are ca. hourly and 1-10 km$^2$. These scales are much smaller than the temporal and spatial scales of alternative techniques for measuring gas transfer, e.g. by dual tracer methods (daily and 1000 km$^2$) (Nightingale et al., 2000; Ho et al., 2006). EC measurements are thus potentially better-suited to capture variations in gas exchange due to small-scale processes at the air-sea interface (Garbe et al., 2014).





The EC $CO_2$ flux method has developed and improved over time. Before 1990, EC was successfully used to measure air-sea momentum and heat fluxes. EC air-sea $CO_2$ flux measurements made during those times were unreasonably high (Jones and Smith, 1977; Wesely et al., 1982; Smith and Jones, 1985; Broecker et al., 1986). After 1990, with the development of the infrared gas analyser, EC became routinely used for terrestrial carbon cycle research (Baldocchi et al., 2001). Development of the EC method was accompanied by improvements in the flux uncertainty analysis, which was generally based on momentum, heat and land-atmosphere gas flux measurements (Lenschow and Kristensen, 1985; Businger, 1986; Lenschow et al., 1994; Wienhold et al., 1995; Mahrt, 1998; Finkelstein and Sims, 2001; Loescher et al., 2006; Rannik et al., 2009, 2016; Billesbach, 2011; Mauder et al., 2013; Langford et al., 2015; Post et al., 2015).

In the late 1990s, the advancement in motion correction of wind measurements (Edson et al., 1998; Yelland et al., 1998) facilitated ship-based EC $CO_2$ flux measurements from a moving platform (McGillis et al., 2001; 2004). After 2000, a commercial open-path infrared gas analyser LI-7500 became widely used for air-sea $CO_2$ flux measurements (Weiss et al., 2007; Kondo and Tsukamoto, 2007; Prytherch et al., 2010; Edson et al., 2011; Else et al., 2011; Lauvset et al., 2011). The LI-7500 generated extremely large and highly variable $CO_2$ fluxes in comparison to expected (Kondo and Tsukamoto, 2007; Prytherch et al., 2010; Edson et al., 2011; Else et al., 2011; Lauvset et al., 2011), which are generally considered to be an artefact caused by water vapour cross-sensitivity (Kohsiek, 2000; Prytherch et al., 2010; Edson et al., 2011; Landwehr et al., 2014). Mathematical corrections proposed to address this artefact (Edson et al., 2011; Prytherch et al., 2010) were later shown to be unsatisfactory (Else et al., 2011; Ikawa et al., 2013; Blomquist et al., 2014; Tsukamoto et al., 2014) or incorrect (Landwehr et al., 2014).

The most reliable method for measuring EC air-sea $CO_2$ fluxes involves physical removal of water vapour fluctuations from the sampled air. The simplest approach is to combine a closed-path gas analyser with a physical dryer to eliminate most of the water vapour fluctuation (Miller et al., 2010; Blomquist et al., 2014; Landwehr et al., 2014; Yang et al., 2016; Nilsson et al., 2018). The tuneable-diode-laser-based cavity ring-down spectrometer (CRDS) made by Picarro Inc. (Santa Clara, California, USA) is the most precise closed-path analyser currently available (Blomquist et al., 2014). The closed-path infrared gas analyser LI-7200 (LI-COR Biosciences, Lincoln, Nebraska, USA) is another popular choice.





The advancements in instrumentation and in motion correction methods have significantly improved the quality of air-sea EC $CO_2$ flux observations but, despite these changes, the flux uncertainties have not been well-quantified. The aims of this study are to: 1) analyse uncertainties in EC air-sea $CO_2$ flux measurements; 2) propose practical methods to reduce the systematic and random flux uncertainty; and 3) investigate how the EC flux uncertainty influences our ability to estimate and parameterise $K_{660}$.

## 2 Experiment and methods

### 2.1 Instrumental set-up

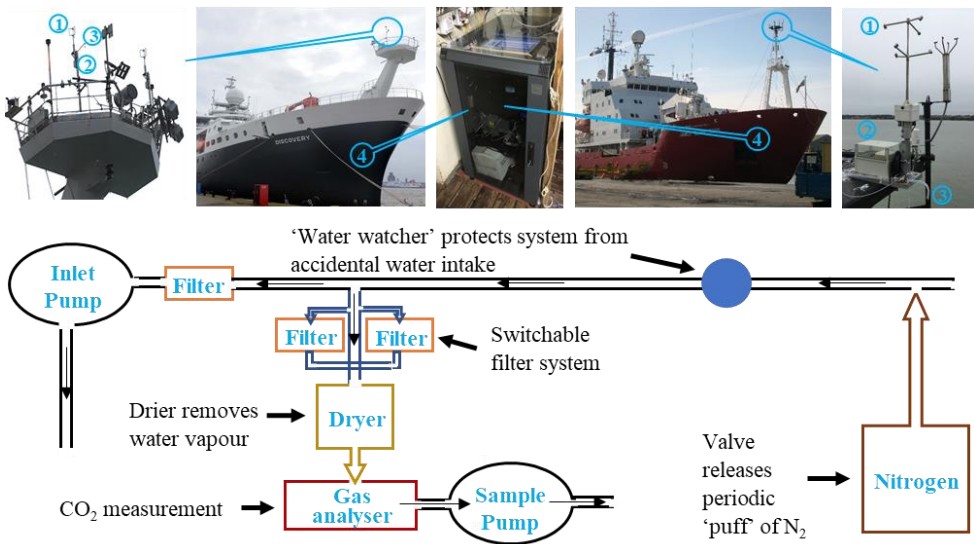

**Figure 1.** EC system (upper panel) and a diagram of system setup (bottom panel). EC instruments: 1) Sonic anemometer, 2) Motion sensor, 3) Air sample inlet for gas analyser, 4) Datalogger/gas analyser. Arctic and Atlantic data from 2018 were collected on the RRS James Clark Ross (JCR, upper right) using a Picarro G2311-f, and Atlantic data from 2019 were collected using a LI-7200 on the RRS Discovery (upper left).

The basic information of four cruises is summarised in Table 1. Appendix A shows the four cruise tracks (Fig. A1, A2). Data from the Atlantic cruises (AMT28 and AMT29) are limited to





3° N–20° S in order to focus specifically on the performance of two different gas analysers in the
same region with low flux signal (tropical zone).

**Table 1.** Basic information for all four cruises on the RRS James Clark Ross (JCR) and RRS Discovery
that measured air-sea EC $CO_2$ fluxes.

| Cruise | JR18006 | JR18007 | AMT28 | AMT29 |
|---|---|---|---|---|
| **Data period** | 30 June–1 August 2019 | 5 August–29 September 2019 | 9 October–16 October 2018 | 4 November–11 November 2019 |
| **Visited region** | Arctic Ocean (Barents Sea) | Arctic Ocean (Fram Strait) | Tropical Atlantic Ocean | Tropical Atlantic Ocean |
| **Research vessel** | JCR | JCR | JCR | Discovery |
| **Gas analyser** | Picarro G2311-f | Picarro G2311-f | Picarro G2311-f | LI-7200 |


The $CO_2$ flux and data logging systems installed on the JCR and Discovery were operated
autonomously. The EC systems were approximately 20 m above mean sea level on both ships
(at the top of the foremasts, Fig. 1) to minimise flow distortion and exposure to sea spray.
Computational fluid dynamics (CFD) simulation indicates that the airflow distortion at the top
of the JCR foremast is small (~1% of the free stream wind speed when the ship is head to wind,
Moat and Yelland, 2015). The hull structure of RRS Discovery is nearly identical to that of
RRS James Cook. CFD simulation of the James Cook indicates that the airflow at the top
foremast is distorted by ~2% for bow-on flows (Moat et al., 2006).
The EC system on the JCR consists of a three-dimensional sonic anemometer (Metek Inc.,
Sonic-3 Scientific), a motion sensor (initially Systron Donner Motionpak II, which compared
favourably with and was then replaced by a Life Performance-Research LPMS-RS232AL2 in
April 2019), and a Picarro G2311-f gas analyser. All instruments sampled at a frequency of 10
Hz or greater and the data were logged at 10 Hz with a datalogger (CR6, Campbell Scientific,
Inc.), similar to the setup by Butterworth and Miller (2016). Air is pulled through a long tube
(30 m, 0.95 cm inner diameter) with a dry vane pump at a flow rate of ~40 L min$^{-1}$ (Gast 1023
series). The Picarro gas analyser subsamples from this tube through a particle filter (Swagelok
2 µm) and a dryer (Nafion PD-200T-24M) at a flow of ~5 L min$^{-1}$ (Fig. 1). The dryer is setup
in the 're-flux' configuration and uses the lower pressure Picarro exhaust to dry the sample air.
This method removes ~80% of the water vapour and essentially all of the humidity fluctuations





(Yang et al., 2016). The Picarro internal calculation accounts for the detected residual water
vapour and yields a dry $CO_2$ mixing ratio that is used in the flux calculations. A valve controlled
by the Picarro instrument injects a 'puff' of nitrogen ($N_2$) into the tip of the inlet tube for 30 s
every 6 hours. This enables estimates of the time delay and high-frequency signal attenuation
(Sect. 2.2).
The EC system on RRS Discovery consists of a Gill R3-50 sonic anemometer, a LPMS motion
sensor package, and a LI-7200 gas analyser. The LI-7200 gas analyser was mounted within the
enclosed staircase, directly underneath the meteorological platform and close to the inlet (inlet
length 7.5 m). A single pump (Gast 1023) was sufficient to pull air through a particle filter
(Swagelok 2 µm), a dryer (Nafion PD-200T-24M), and the LI-7200 at a flow of ~7 L min$^{-1}$.
There was no $N_2$ puff system setup on Discovery but equivalent lab tests confirmed that the
delay time was less than on the JCR because of the shorter inlet line. The dryer on the Discovery
is setup in the same 're-flux' configuration as the JCR and uses the lower pressure at the LI-
7200 exhaust (limited by an additional 0.08 cm diameter critical orifice) to dry the sample air.
This setup removes ~60–70% of the water vapour and essentially all of the humidity
fluctuations. The dry $CO_2$ mixing ratio, computed by accounting for the LI-7200 temperature,
pressure and residual water vapour measurements, is used in the flux calculations.

**2.2 Flux processing**


The EC air-sea $CO_2$ flux calculation steps using the raw data are outlined with a flow chart
(Fig. 2) and detailed below. The raw high frequency wind and $CO_2$ data are processed first,
yielding fluxes in 20 min averaging time interval and related statistics. These statistics are then
used for quality control of the fluxes. Further averaging of the quality-controlled 20 min fluxes
to hourly or longer time scales is then used to reduce random error (Sect. 4.1). Linear
detrending was used to identify the turbulent fluctuations (i.e. $w'$ and $c'$) throughout the
analyses.
To correct the wind data for ship motion, we first generated hourly data files containing the
measurements from the sonic anemometer (three-dimensional wind speed components: $u$, $v$
and $w$ and sonic temperature $Ts$), motion sensor (three axis accelerations: accel_x, accel_y,
accel_z; and rotation angles: rot_x, rot_y, rot_z ), ship heading over ground (HDG, from the
gyro compass) and ship speed over ground (SOG, from Global Position System). Spikes larger
than 4 standard deviations (SDs) from the median were removed. Secondly, a complementary
filtering method using Euler angles (see Edson et al., 1998) was applied to the hourly data files



to remove apparent winds generated by the ship movements. The motion-corrected winds were
further decorrelated against ship motion to remove any residual motion-sensitivity (Miller et
al., 2010; Yang et al., 2013). The motion-corrected winds were double rotated to account for
the wind streamline over the ship, yielding the vertical wind velocity ($w$) required in Eq. 2.
Inspection of frequency spectra showed that the spectral peak at the ship motion frequencies
(approximately 0.1−0.3 Hz) had disappeared after the motion correction (Fig. S1, Supplement).
This indicates that the majority of ship motion had been removed from the measured wind
speed. The last step in the wind data processing was the calculation of 20 min average friction
velocity, sensible heat flux and other key variables used for data quality control (Table S1,
Supplement).
The $CO_2$ data were de-spiked (by removing values > 4 SDs from the median). The Picarro $CO_2$
mixing ratio was further decorrelated against analyser cell pressure and temperature to remove
$CO_2$ variations due to ship's motion. The LI-7200 $CO_2$ mixing ratio was further decorrelated
against the LI-7200 $H_2O$ mixing ratio and temperature to remove residual air density
fluctuations, following Landwehr et al. (2018). $CO_2$ data were also decorrelated against ship's
heave and accelerations because these can produce spurious $CO_2$ variability (Miller et al., 2010;
Blomquist et al., 2014).
A lag between $CO_2$ data acquisition and the wind data is created because of the time taken for
sample air to travel through the inlet tube. On the JCR, we use the 'puff' system where the lag
time is the time difference between the $N_2$ 'puff' start (when the on/off valve is switched) and
the time when the diluted signal is sensed by the gas analyser. The lag time can also be
estimated by the maximum covariance method, calculated by shifting the time base of the $CO_2$
signal and finding the shift that achieves maximum covariance between the vertical wind
velocity ($w$) signal and the shifted $CO_2$ signal. The lag times estimated by the maximum
covariance method agree well with the estimates of the 'puff' procedure (Fig. S2, Supplement).
These estimates indicate a lag time of 3.3–3.4 s for the Arctic cruises and 3.3 s for cruise
AMT28 on the JCR. The maximum covariance method estimated lag time on Discovery
(AMT29) was 2.6 s, consistent with laboratory test results prior to the cruise.



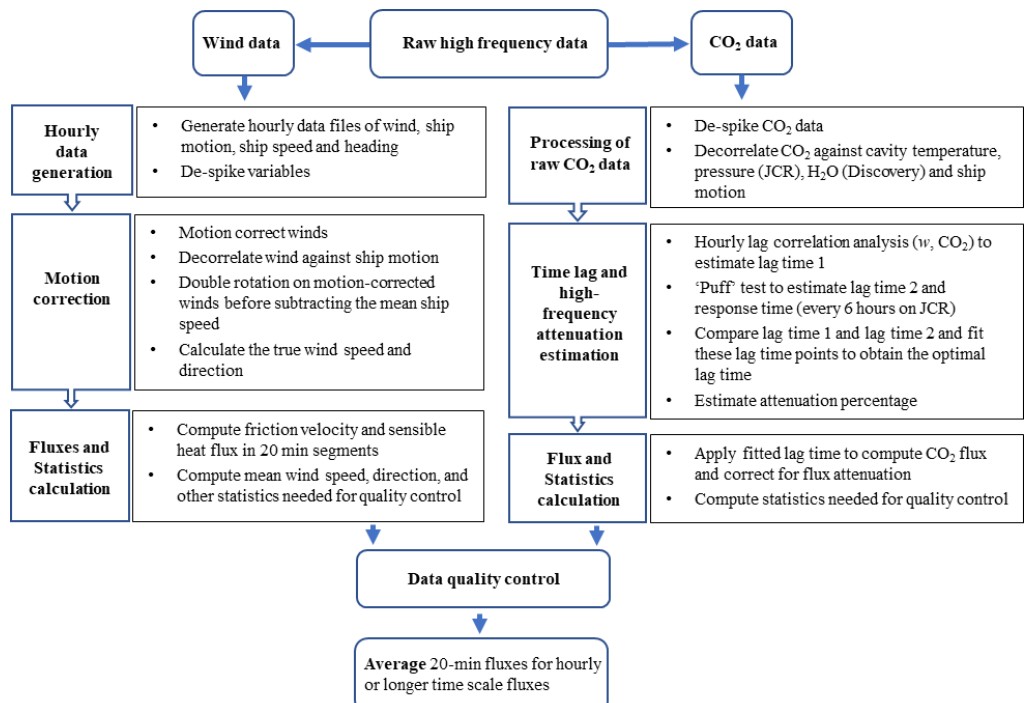


**Figure 2.** Flow chart of EC data processing. The raw high frequency (10 Hz) wind and $CO_2$ data were initially processed separately and then combined to calculate fluxes. $CO_2$ fluxes were filtered by a series of data quality control criteria. The 20-min flux intervals were averaged to longer time scales (hourly or more). The data processing is detailed in the text.

204

The inlet tube, particle filter and dryer cause high-frequency $CO_2$ flux signal attenuation. The $N_2$ 'puff' was also used to assess the response time by considering the e-folding time in the $CO_2$ signal change (similar approaches have been used by Bariteau et al., 2010; Blomquist et al., 2014, Bell et al., 2015). The response time is 0.35 s for the EC system on JCR and 0.25 s for the EC system on Discovery (estimated in the laboratory prior to cruise). These response times were combined with the relative wind speed-dependent, theoretical shapes of the cospectra (Kaimal et al., 1972) to estimate the percentage flux loss due to the inlet attenuation (Yang et al., 2013). The mean attenuation percentage is less than 10% with a relative wind speed dependence (Fig. S3, Supplement). The attenuation percentage value was applied to the computed flux to compensate the flux loss due to the high-frequency signal attenuation. Finally, horizontal $CO_2$ fluxes and other statistics such as $CO_2$ range and $CO_2$ trend were computed for quality control purposes (Table S1, Supplement).





The computed 20-min fluxes were filtered for non-ideal ship manoeuvres or violations of the
homogeneity/stationary requirement of EC (see Supplement for the quality control criteria).

**2.3 Uncertainty analysis methods**

**2.3.1 Uncertainty components**

Uncertainty contains two components: systematic error ($\delta F_S$) and random error ($\delta F_R$).
According to propagation of uncertainty theory (JCGM, 2008), the total uncertainty in EC $CO_2$
fluxes (from random and systematic errors) can be expressed as:

$$\delta F = \sqrt{\delta F_R{}^2 + \delta F_S{}^2} \tag{3}$$

Systematic errors (Sect. 2.3.2) will cause bias in the flux. They thus should be
eliminated/minimised with appropriate system setup and, if needed, effective numerical
corrections. Random error results in imprecision (but not bias) and can be reduced by averaging
repeated measurements (Sect. 2.3.3). Errors due to insufficient sampling and instrument noise
are generally considered most important in EC flux measurements (Lenschow and Kristensen,
1985; Businger 1986; Mauder et al., 2013; Rannik et al., 2016).
Sampling error is an inherent issue for EC flux measurements and is typically the main source
of the $CO_2$ flux uncertainty (Mauder et al., 2013). The sampling error is caused by the
difference between the ensemble average and the time average. The calculation of EC flux (Eq.
2) requires the separation between the mean and fluctuating components, which can be
represented fully for $CO_2$ mixing ratio $c$ as:

$$c(x,t) = \bar{c}(x,t) + c'(x,t) \tag{4}$$

The mean component $\bar{c}$ represents ensemble average over time ($t$) and space ($x$) and does not
contribute to the flux. The time average of a stationary turbulent signal and space average of a
homogenous turbulent signal theoretically converge on the ensemble average when the
averaging time approaches infinity, i.e. $T \rightarrow \infty$ (Wyngaard, 2010). In practice, Reynolds
averaging over a much shorter time interval (10 min to an hour) is typically used for EC flux
measurements from a fixed point or from a slow-moving platform such as a ship. This is
because the atmospheric boundary layer is only quasi-stationary for a few hours. Non-
stationarity (e.g. diurnal variability and synoptic conditions) is an inherent property of the





atmospheric boundary layer (Wyngaard, 2010). EC flux obervations thus inevitably contain
some random error due to insufficient samping time, and this error is greater at shorter
averaging times.
Random error due to instrument noise comes mainly from the white noise of the gas analyser,
as the noise from the sonic anemometer is relatively unimportant (Blomquist et al., 2010;
Fairall et al., 2000; Mauder et al., 2013). Blomquist et al. (2014) show 'pink' noise with a weak
spectral slope for their CRDS gas analyser (G1301-f), but the gas analysers on JCR (G2311-f)
and Discovery (LI-7200) demonstrate white noise with a constant variance at high frequency
(Fig. B2, Appendix B).

**2.3.2 Systematic error**
Table 2 details the measures taken during instrument setup and data processing that help
eliminate most sources of systematic error in EC $CO_2$ fluxes.

**Table 2.** Potential sources of bias in our EC air-sea $CO_2$ flux measurements and the methods used to
minimise them.

| Potential source of bias | Methods used to minimise the bias | Flux uncertainty |
|---|---|---|
| $\delta F_{S,1}$ **Water vapour cross-sensitivity** | Closed-path gas analyser with a dryer removes essentially all of the water vapour fluctuation (Blomquist et al., 2014; Yang et al., 2016). The residual $H_2O$ signal is measured by the gas analyser and used in the calculation of dry $CO_2$ mixing ratio, which removes water cross-sensitivity. | Negligible |
| $\delta F_{S,2}$ **Ship motion** | Flux uncertainty from an earlier version of the motion correction procedure (less rigorous than the one used by ourselves) is estimated to be 10-20% (Edson et al. 1998). The more recently-adopted decorrelation of vertical winds and $CO_2$ against platform motion (Miller et al., 2010; Yang et al., 2013) reduces this uncertainty. Flügge et al. (2016) compare EC momentum fluxes measured from a moving platform (buoy) with fluxes measured from a nearby fixed tower. Flux estimates from these two platforms agree well (relative flux bias due to the motion correction $\leq 6\%$). | $\leq 6\%$ |





| $\delta F_{S,3}$ **Airflow distortion** | The EC flux system is deployed as far forward and as high as possible on the ship (top of the foremast), which minimises the impacts of flow distortion. Subsequent distortion correction using the CFD simulation (Moat et al., 2006; Moat and Yelland, 2015) along with a relative wind direction restriction further reduces the impact of flow distortion on the fluxes. Measured EC friction velocities and friction velocities from the COARE3.5 model (Edson et al., 2013) agree well (e.g. $R^2 = 0.95$, slope = 0.97) for data collected during cruise JR18006. Good comparison between observed and COARE3.5 friction velocity estimates indicates that we have fully accounted for flow distortion effects. | Negligible |
|---|---|---|
| $\delta F_{S,4}$ **Inlet effects (high-frequency flux attenuation and $CO_2$ sampling delay)** | High-frequency flux signal attenuation (in the inlet tube, particle filter and dryer) is evaluated by the $CO_2$ signal response to a puff of $N_2$ gas. Flux attenuation is calculated from the 'inlet puff' response and applied as a correction (< 10%, see Sect. 2.2). The uncertainty in the attenuation correction is about 1% for unstable/neutral atmospheric conditions, which is generally the case over the ocean (e.g. 93% of the time for the Atlantic cruises, 80% of the time for the Arctic cruises). During stable conditions, the attenuation correction is larger (Landwehr et al., 2018) and the uncertainty is also greater (~20%).<br><br>The lag time adjustment prior to the flux calculation aligns the $CO_2$ and wind signals. Two methods are used to estimate the optimal lag time: puff injection and maximum covariance. The two lag estimates are in good agreement (Sect. 2.2). Random adjustment of ± 0.2 s (1 σ of the puff test result) to the optimal lag time impacts the $CO_2$ flux by < 1%. | < 2% for vast majority of the cruises |
| $\delta F_{S,5}$ **Spatial separation between the sonic anemometer and the gas inlet** | The $CO_2$ inlet is ~70 cm directly below the centre volume of the sonic anemometer. This distance is small relative to the size of the dominant flux-carrying eddies encountered by the EC measurement system height above sea level. The excellent agreement between the lag time determined by the puff system and by the optimal covariance method further confirms that the distance between the $CO_2$ inlet and anemometer is sufficiently small. | Negligible |





| $\delta F_{S,6}$ **Imperfect calibration of the sensors** | The potential flux bias resulting from instrument calibration (gas analyser, anemometer and meteorological sensors required to calculate air density: air temperature, relative humidity and pressure) is up to 4% for the JCR setup. The largest instrument calibration uncertainty derives from the wind sensor accuracy ($\pm$ 0.15 m s$^{-1}$ at 4 m s$^{-1}$ winds according to the Metek uSonic instrument specification). This bias is even lower ($< 2\%$) for the Discovery setup because the Gill R3 sonic anemometer is more accurate. | $\leq 4\%$ |
|---|---|---|
| **Propagated bias** | Estimated from the individual bias estimates above ($\delta F_{S,1}, \delta F_{S,2}$, etc.) using $\delta F_S = \sqrt{\sum_1^n \delta F_{S,n}^2}$ | $< 7.5\%$ |


In addition to bias sources related to the instrument setup (Table 2), insufficient sampling time
(an inherent issue of EC fluxes) may also generate a systematic error. We use a theoretical
method to estimate this systematic error in EC $CO_2$ flux (Lenschow et al., 1994):

$$|\delta F_S| \leq 2\sigma_w \sigma_{c_a} \frac{\sqrt{\tau_w \tau_c}}{T} \tag{5}$$

where $\sigma_w$ (m s$^{-1}$) and $\sigma_{c_a}$ (ppm) are the standard deviations of the vertical wind velocity and
the $CO_2$ mixing ratio due to atmospheric processes, respectively. $T$ is the averaging time
interval (s), and $\tau_w$ and $\tau_c$ are integral time scales (s) for vertical wind velocity and $CO_2$ signal,
respectively. The definition and estimation of the integral time scale are shown in Appendix B.
The sign of $\delta F_S$ could be positive or negative (i.e. under or over-estimation) because of the
poor statistics in capturing low-frequency eddies within the flux averaging period (Lenschow
et al., 1993). The mean hourly relative systematic error due to insufficient sampling time for
four cruises estimated by Eq. 5 is $< 5\%$. According to propagation of uncertainty theory (JCGM,
2008), the total systematic error is less than 9% ($= \sqrt{7.5\%^2 + 5\%^2}$).

### 2.3.3 Random error

Five approaches used to estimate the total random error (A-C) and the random error component
due to instrument noise (C-E) in EC $CO_2$ fluxes are discussed below. The random error
assessments are empirical (A and D) or theoretical (B, C and E).

**A.** An empirical approach to estimate total random error involves shifting the *w* data relative
to the $CO_2$ data (or vice versa) by a large, unrealistic time shift and then computing the 'null





fluxes' from the time-desynchronized $CO_2$ and $w$ time series (Rannik et al., 2016). The shift
removes any real correlation between $CO_2$ and $w$ due to vertical exchange. The standard
deviation of the resultant 'null' fluxes represents the random flux uncertainty (Wienhold et al.,
1995). We applied a series of time shifts of $\sim 20 - 60 \times \tau_w$ (i.e. using time shifts ranging from
-300 to -100 and 100 to 300 s, Rannik et al., 2016). This empirical estimation of total random
flux uncertainty will hereafter be referred to as $\delta F_{R,\text{Wienhold}}$.
**B.** Lenschow and Kristensen (1985) derived a rigorous theoretical equation for total random
error estimation, which contains both the auto-covariance and cross-covariance functions. The
theoretical equation has been numerically approximated by Finkelstein and Sims (2001):
$$\delta F_{R,\text{ Finkelstein}} = \left\{ \frac{1}{n} \left[ \sum_{p=-m}^{m} r_{ww}(p) r_{cc}(p) + \sum_{p=-m}^{m} r_{wc}(p) r_{cw}(p) \right] \right\}^{1/2} \qquad (6)$$

where $n$ is the number of data points within an averaging time interval, $p$ is the number of
shifting points. The maximum shifting point $m$ can be chosen subjectively $(< n)$. We found that
the random error for $m$ between 1000 and 2000 data points was similar, so for this study we
use $m = 1500$ (150 s shift time). The first term in the brackets represents the auto-covariance
component and the second term is the cross-covariance component. $r_{ww}$ and $r_{cc}$ are the auto-
covariance functions for vertical wind velocity ($w$) and $CO_2$ mixing ratio ($c$), respectively. $r_{wc}$
and $r_{cw}$ are the cross-covariance functions for $w$ and $c$. Here $r_{wc}$ represents shifting $w$ data
relative to $CO_2$ data, while $r_{cw}$ represents shifting $CO_2$ data relative to $w$ data.
**C.** Blomquist et al. (2010) attributed the sources of $CO_2$ variance $\sigma_c^2$ to atmospheric processes
($\sigma_{c_a}^2$) and white noise ($\sigma_{c_n}^2$). The sources of variance are considered to be independent of each
other and the sonic anemometer is assumed to be relatively noise-free. According to
propagation of uncertainty theory (JCGM, 2008), the total random flux error can be defined as:
$$\delta F_{R,\text{ Blomquist}} \le \frac{a \sigma_w}{\sqrt{T}} \left( \sigma_{c_a}^2 \tau_{wc} + \sigma_{c_n}^2 \tau_{c_n} \right)^{1/2} \qquad (7)$$

where the constant $a$ varies from $\sqrt{2}$ to 2, depending on the relationship between the covariance
of the two variables ($w$ and $CO_2$) and the product of their auto-correlations (Lenschow and
Kristensen, 1985). Here, $\tau_{wc}$ is equal to the shorter of $\tau_w$ and $\tau_c$, which is typically $\tau_w$
(Blomquist et al., 2010), and $\tau_{c_n}$ is the integral time scale of white noise in the $CO_2$ signal. The
$CO_2$ variance due to atmospheric processes ($\sigma_{c_a}^2$) includes two components: variance due to
vertical flux (i.e. air-sea $CO_2$ flux) $\sigma_{c_{av}}^2$, and variance due to other atmospheric processes $\sigma_{c_{ao}}^2$



(Fairall et al., 2000). The variance in $CO_2$ due to vertical flux ($\sigma_{c_{av}}^2$) depends on atmospheric
stability. $\sigma_{c_{av}}^2$ can be estimated with Monin-Obukhov similarity theory (Blomquist et al., 2010,
2014; Fairall et al., 2000):

$$\sigma_{c_{av}}^2 = \left[3\frac{\overline{w'c'}}{u_*}f_c(z/L)\right]^2 \tag{8}$$


where $u_*$ is the friction velocity (m s$^{-1}$) and the similarity function ($f_c$) depends on the stability
parameter $z/L$, where $z$ is the observational height (m) and $L$ is the Obukhov length (m). The
expression of $f_c$ can be found in Blomquist et al. (2010).
Equation 7 can be used to assess the random error due to instrument noise by setting $\sigma_{c_a}^2 = 0$,
referred to hereafter as $\delta F_{RN,\,Blomquist}$. We use the $CO_2$ variance spectra to directly estimate the
white noise term $\sigma_{c_n}^2 \tau_{c_n}$ in Eq. 7. The variance is fairly constant at high frequency (1-5 Hz; Fig.
B2, Appendix B), which is often referred to as band-limited white noise. The relationship
between $\sigma_{c_n}^2 \tau_{c_n}$ and the band-limited noise spectral value $\varphi_{c_n}$, is expressed in Blomquist et al.
(2010) as:

$$\sigma_{c_n}^2 \tau_{c_n} = \frac{\varphi_{c_n}}{4} \tag{9}$$


**D.** Billesbach (2011) developed an empirical method to estimate the random error due to
instrument noise alone (referred to as $\Delta F_{RN,\,Billesbach}$). This involves random shuffling of the
$CO_2$ time series within an averaging interval and then calculating the covariance of $w$ and $CO_2$.
The correlation between $w$ and $CO_2$ is minimized by the shuffling, and any remaining
correlation between $w$ and $CO_2$ is due to the unintentional correlations contributed by
instrument noise.
**E.** Mauder et al. (2013) describe another theoretical approach to estimate the random flux error
due to instrument noise:

$$\delta F_{RN,\,Mauder} = \frac{\sigma_w \sigma_{c_n}}{\sqrt{n}} \tag{10}$$


White noise correlates with itself but is uncorrelated with atmospheric turbulence. Thus, the
white noise-induced $CO_2$ variance ($\sigma_{c_n}$) only contributes to the total variance. The value of $\sigma_{c_n}$
can be estimated from the difference between the zero-shift auto-covariance value ($CO_2$
variance $\sigma_c^2$) and the noise-free variance extrapolated to a time shift of zero (Lenschow et al.,

338 2000):





$$\sigma_{c_n}^2 = \sigma_c^2 - \sigma^2(t \to 0) \tag{11}$$

where $\sigma^2(t \to 0)$ represents the extrapolation of auto-covariance to a zero shift, which is
considered equal to variance due to atmospheric processes ($\sigma_{c_a}^2$). Figure 3 shows the normalised
auto-covariance function curves of $w$ and $CO_2$ as measured by the Picarro G2311-f and the LI-
7200. There is a sharp decrease in the $CO_2$ auto-covariance when shifting from 0 s shift to 0.1
s shift for both the Picarro G2311-f and LI-7200 gas analyser. The same sharp decrease is not
seen in the vertical wind velocity ($w$) auto-covariance. The relative difference in the change in
normalised auto-covariance shows that white noise makes a much larger relative contribution
to the $CO_2$ variance than to the vertical wind velocity variance.

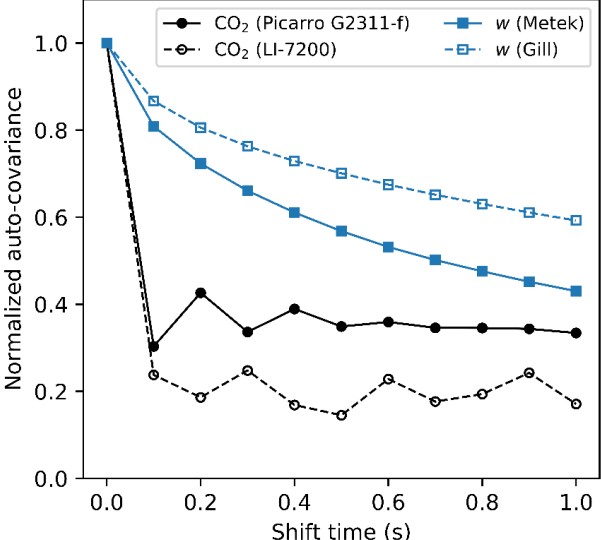


**Figure 3.** Mean normalised auto-covariance functions of $CO_2$ and vertical wind velocity ($w$) by four
different instruments. The sharp decrease of the $CO_2$ auto-covariance between the zero shift and the
initial 0.1 s shift corresponds to the large contribution of white noise from the gas analysers. The LI-
7200 is the nosier instrument. The noise contribution from either anemometer is relatively small ($<$

353 10%).


## 3   Results

Measurements from AMT28 and AMT29 set the scene for our uncertainty analysis. These two
Atlantic cruises transited across the same tropical region (Fig. A2, Appendix A) in October





2018 and September 2019 with different eddy covariance systems (Sect. 2.1). AMT28 and
AMT29 show broadly similar latitudinal patterns (Fig. 4a). An obvious question of interest is
whether the measured fluxes were the same for the two years. To answer this question, the
measurement uncertainties must be quantified. The total random uncertainties in $CO_2$ flux
($\delta F_{R, \text{Finkelstein}}$) are comparable for the two cruises even though the random error component
due to instrument noise ($\delta F_{RN, \text{Mauder}}$) is about 3 times higher during AMT29 using LI-7200
than during AMT28 using Picarro G2311-f (Fig. 4b; Fig. D1, Appendix D). The similar total
random uncertainty in the AMT28 and AMT29 fluxes shows that both gas analysers are equally
suitable for air-sea EC $CO_2$ flux measurements. The variance budgets of atmospheric $CO_2$
mixing ratio (used to estimate random flux uncertainty, see Sect. 3.1) are shown in Fig. 4c.
Total variance in $CO_2$ mixing ratio is dominated by instrument noise on both cruises. $CO_2$
mixing ratio variance (total and instrument noise) was substantially higher during AMT29.

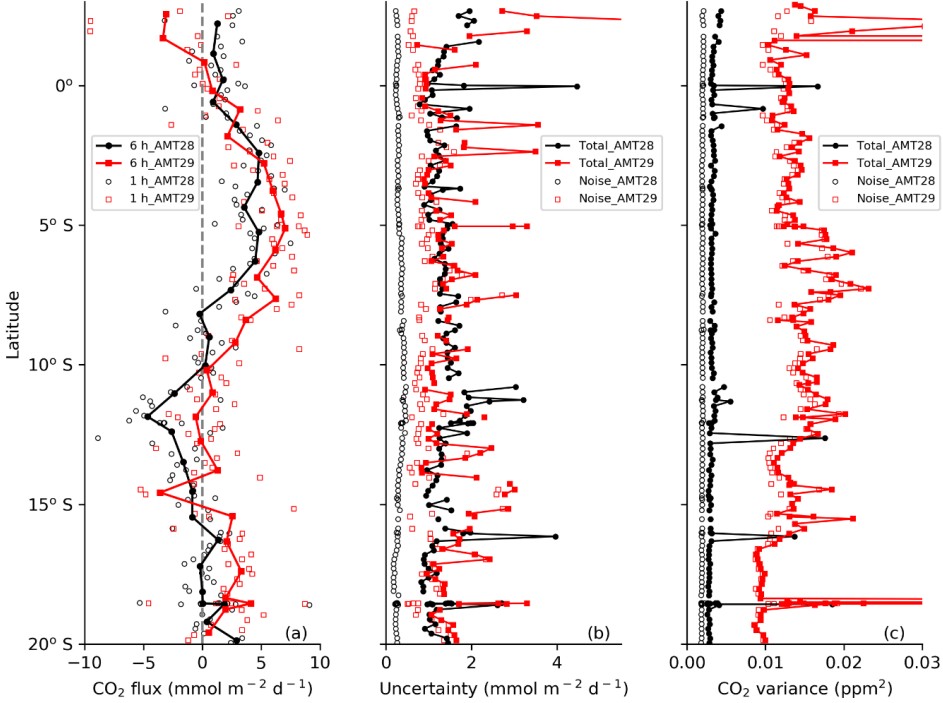


**Figure 4.** (a) Air-sea $CO_2$ fluxes (hourly and 6-h averages), (b) random uncertainty in flux (total and
due to instrument noise only), and (c) variance in $CO_2$ mixing ratio (total and due to instrument noise
only) for two Atlantic cruises.





### 3.1 Random uncertainty

Theoretical derivation of flux uncertainty ($\delta F_{RN, \text{Blomquist}}$, Eq. 7) requires knowledge of the contributions to $CO_2$ mixing ratio variance. Total $CO_2$ variance is made up of instrument noise ($\sigma_{c_n}^2$) and atmospheric processes ($\sigma_{c_a}^2$). Atmospheric processes include vertical flux ($\sigma_{c_{av}}^2$) and other atmospheric processes ($\sigma_{c_{ao}}^2$). The variance budgets of $CO_2$ mixing ratio for the four cruises are listed in Table 3. Atmospheric processes contribute a larger $CO_2$ variance in the Arctic (where flux magnitudes are greater) compared to the Atlantic. Vertical flux accounts for ~10% of the variance in $CO_2$ mixing ratio in the Arctic and ~1% of the $CO_2$ variance in the Atlantic. Previous results demonstrate that horizontal transport is a major source of $\sigma_{c_{ao}}^2$ for long-lived greenhouse gases (Blomquist et al., 2012). Small changes in $CO_2$ mixing ratio transported horizontally can yield variance that greatly exceeds the variance from vertical flux.

**Table 3.** Variance in the $CO_2$ mixing ratio estimated using Eq. 8 and 11 for the Arctic (JR18006/7, Picarro G2311-f) and Atlantic cruises (AMT28, Picarro G2311-f; AMT29, LI-7200). Total $CO_2$ variance ($\sigma_c^2$) consists of white noise ($\sigma_{c_n}^2$) and atmospheric processes ($\sigma_{c_a}^2$). The latter can be further broken down to the $CO_2$ variance due to vertical flux ($\sigma_{c_{av}}^2$) and due to other processes ($\sigma_{c_{ao}}^2$).

| $CO_2$ variance ($\times\ 10^{-3}$ ppm$^2$) | JR18006 | JR18007 | AMT28 | AMT29 |
|---|---|---|---|---|
| **Total, $\sigma_c^2$** | 9.9 | 8.6 | 3.6 | 13.9 |
| **Due to instrument white noise, $\sigma_{c_n}^2$** | 5.8 | 5.4 | 2.0 | 12.6 |
| **Due to atmospheric processes, $\sigma_{c_a}^2$** | 4.1 | 3.3 | 1.6 | 1.3 |
| **- Due to vertical flux, $\sigma_{c_{av}}^2$** | 1.3 | 0.8 | 0.03 | 0.08 |
| **- Due to other atmospheric processes, $\sigma_{c_{ao}}^2$** | 2.8 | 2.5 | 1.6 | 1.2 |

Three quasi-independent methods were used to estimate random uncertainty in EC air-sea $CO_2$ fluxes caused by instrument noise ($\delta F_{RN}$, Methods C-E, Sect. 2.3.3). Good agreement was found between all three estimates (Fig. C2, Appendix C) when $\sqrt{2}$ is used as the constant in Eq. 7 ($a$). The $\Delta F_{RN, \text{Billesbach}}$ estimates have more scatter and are slightly higher than the theoretical results, possibly because the random shuffling of data fails to fully exclude the contribution from atmospheric turbulence (Rannik et al., 2016). For the remainder of this study, we use the $\delta F_{RN, \text{Mauder}}$ method to estimate $\delta F_{RN}$.

We used three methods to estimate the total random uncertainty ($\delta F_R$, Methods A-C, Sect. 2.3.3) in the hourly-averaged air-sea $CO_2$ fluxes. There is good agreement among the three estimates



(r > 0.88; Fig. C1, Appendix C). Again, the constant in Eq. 7 (*a*) is set to $\sqrt{2}$, as informed by
the instrument noise uncertainty analysis above. We use $\delta F_{R,\,Finkelstein}$ (Eq. 6) to estimate the
total random flux uncertainty hereafter. Our decision is based on $\delta F_{R,\,Finkelstein}$ not requiring
the integral time scale (unlike $\delta F_{R,\,Blomquist}$) and showing less scatter than $\delta F_{R,\,Wienhold}$.
Figure 5 shows the different relative contributions to the random flux uncertainty for the Arctic
cruises (hourly average). Here the uncertainty is normalised by the flux magnitude and then
averaged into flux magnitude bins. When the flux magnitude is sufficiently large (> 20 mmol
m$^{-2}$ day$^{-1}$), the total relative random uncertainty in flux asymptotes to about 15% and is driven
by variance associated with both vertical flux and other atmospheric processes. This estimate
is similar to uncertainties in air-sea fluxes of other well resolved (i.e. high signal-to-noise ratio)
variables (Fairall et al., 2000). At a lower flux magnitude, uncertainty due to atmospheric
processes other than vertical flux dominates the total random uncertainty. Uncertainty due to
the white noise from the Picarro G2311-f gas analyser is small.

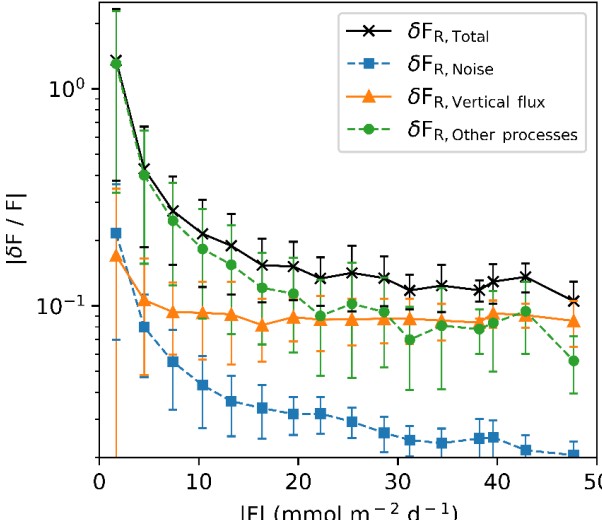


**Figure 5.** Relative random uncertainty in hourly CO$_2$ flux and its contribution from noise, vertical flux
and other processes during two Arctic cruises. Relative random uncertainty data are binned into 3 mmol
m$^{-2}$ day$^{-1}$ flux magnitude bins (error bars represent 1 standard deviation).

**3.2 Summary of systematic and random uncertainties**





The total uncertainty $\delta F$ in the hourly average EC $CO_2$ flux (estimated using Eq. 3) ranges
from 1.4 to 3.2 mmol m$^{-2}$ day$^{-1}$ in the mean for the four cruises (Table 4). Our EC flux system
setup was optimal and subsequent corrections have minimised any bias to < 9% (Sect. 2.3.2).
Systematic error is on average much lower than random error (Table 4). This means the
accuracy of the EC $CO_2$ flux measurements is very high, but the precision of hourly averaged
EC $CO_2$ air-sea flux measurements is relatively low. In Sect. 4.1, we discuss how the precision
can be improved by averaging the observed fluxes for longer.

**Table 4.** Summary of hourly average EC $CO_2$ fluxes and associated uncertainties in the mean for the
four cruises (mmol m$^{-2}$ day$^{-1}$). Shown are the mean $CO_2$ flux magnitude ($\overline{|F|}$, mmol m$^{-2}$ day$^{-1}$), upper
limitation of the total uncertainty ($\delta F$, Eq. 3), upper limitation of the absolute systematic error ($|\delta F_S|$,
propagated from Table 2 and Eq. 5), and random error ($\delta F_R$, Eq. 6). The random error components are
white noise ($\delta F_{RN}$, Eq. 10), vertical flux ($\delta F_{RV}$, Eq. 7) and other atmospheric processes ($\delta F_{RO} =$
$\sqrt{\delta F_R^2 - \delta F_{RN}^2 - \delta F_{RV}^2}$). The total uncertainty is also expressed as a % of the mean flux magnitude
($\delta F/|F| \times 100\%$).

| Cruises | JR18006 | JR18007 | AMT28 | AMT29 |
|---|---|---|---|---|
| **$\overline{\lvert CO2\ \mathbf{flux}\rvert}$, $\lvert F\rvert$** | 10.1 | 16.3 | 2.5 | 3.5 |
| **Total uncertainty, $\delta F$** | 2.3 | 3.2 | 1.4 | 1.7 |
| **($\delta F/\lvert F\rvert \times 100\%$)** | (23%) | (20%) | (58%) | (49%) |
| **Systematic error, $\lvert \delta F_S \rvert$** | 0.8 | 1.2 | 0.3 | 0.3 |
| **Total random error, $\delta F_R$** | 2.2 | 2.9 | 1.4 | 1.7 |
| **Random error due to white noise, $\delta F_{RN}$** | 0.5 | 0.6 | 0.3 | 1.0 |
| **Random error due to vertical flux, $\delta F_{RV}$** | 1.1 | 1.4 | 0.2 | 0.4 |
| **Random error due to other atmospheric processes, $\delta F_{RO}$** | 1.5 | 2.4 | 1.4 | 1.5 |


The theoretical uncertainty estimates above can be compared with a portion of the AMT28
cruise data (15°−20° S, ~25° W; Fig. 4), when the ship encountered sea surface $CO_2$ fugacity
close to equilibrium with the atmosphere (i.e. $\Delta f CO_2 \sim 0$, Fig. A2, Appendix A). The data from
this region is useful for assessing the random and systematic flux uncertainties. The standard
deviation of the EC $CO_2$ flux during cruise AMT28 when $\Delta f CO_2 \sim 0$ is 1.6 mmol m$^{-2}$ day$^{-1}$,
which compares well with the theoretical random flux uncertainty in this region (1.4 mmol m$^{-}$
$^2$ day$^{-1}$). The mean EC $CO_2$ flux from this region was 0.5 mmol m$^{-2}$ day$^{-1}$, which is





indistinguishable from zero considering the random uncertainty. This further confirms the
minimal bias in our flux observations.
Figure 6 shows a comparison between the relative uncertainty and the relative standard
deviation (RSTD) in in the hourly $CO_2$ flux for the two Arctic cruises. Results have been binned
into 1 m s$^{-1}$ wind speed bins. Wind speed was converted to 10-meter neutral wind speed ($U_{10N}$)
using the COARE3.5 model (Edson et al., 2013). The relative random error decreases with
increasing wind speed. This is partly because the fluxes tend to be larger at higher wind speeds
and so the signal-to-noise ratio in the flux is greater. In addition, at higher wind speeds, a greater
number of high-frequency turbulent eddies are sampled by the EC system, providing better
statistics of turbulent eddies, and lower sampling error.

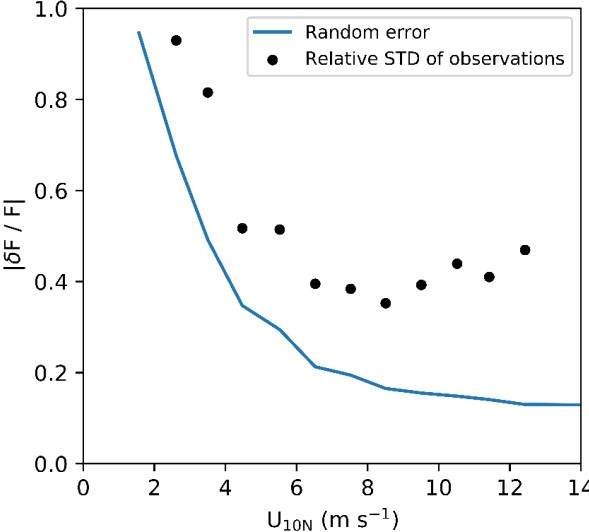


**Figure 6.** Comparison of relative random uncertainty in hourly $CO_2$ flux and relative standard deviation
(RSTD, standard deviation/|flux mean|) of the EC $CO_2$ flux from two Arctic cruises. These results
are binned in 1 m s$^{-1}$ wind speed bins.

The RSTD of the flux is greater in magnitude than the estimated flux uncertainty because it
also contains environmental variability. The $CO_2$ flux auto-covariance analysis (Sect. 4.1)
shows that random error in hourly flux explains ~20% of the flux variance on average for the
two Arctic cruises. This implies that the remaining variability in the EC flux (~80%) is due to
natural phenomena (e.g. changes in $\Delta fCO_2$, wind speed, etc). Similarly, substantial variability



is typical in EC-derived $CO_2$ gas transfer velocity at a given wind speed (e.g. Edson et al., 2011;
Butterworth and Miller, 2016). $K_{660}$ is derived from $(\text{EC } CO_2 \text{ flux})/\Delta f CO_2$, and thus an
understanding of EC flux uncertainty can help understand and explain the variability in EC-
derived gas transfer velocity estimates (Sect. 4.2).
## 4  Discussion
### 4.1 Impact of averaging time scale on flux uncertainty
The random error in flux decreases with increasing averaging time interval $T$ or the number of
sampling points $n$ (Eq. 6, 7 and 10). This is because a longer averaging time interval results in
better statistics of the turbulent eddies. However, averaging for too long is also not ideal since
the atmosphere is less likely to maintain stationarity. The typical averaging time interval is thus
typically between 10 min and 60 min for air-sea flux measurements (20 min intervals were
used in this study). The timeseries of quality controlled 20 min flux intervals can be further
averaged over a longer time scale to reduce the random uncertainty. Averaging the 20 min flux
intervals assumes that the flux interval data are essentially repeat measurements within a
chosen averaging time scale. If the 20 min flux intervals are averaged, one can ask: What is the
optimal averaging time scale for interpreting air-sea EC $CO_2$ fluxes?
We use an auto-covariance method to determine the optimal averaging time scale. The observed
variance in $CO_2$ flux consists of random uncertainty (random noise) as well as natural
variability. The random noise component should only contribute to the $CO_2$ flux variance when
the data are zero-shifted. After the $CO_2$ flux data are shifted, the noise will not contribute to the
auto-covariance function. Figure 7 shows the auto-covariance function of the air-sea $CO_2$ flux
with different averaging time scales for Arctic cruise JR18007. For the 20-min fluxes (Fig. 7a),
the auto-covariance decreases rapidly between the zero shift and the initial time shift, which
indicates that a large fraction of the 20-min flux variance is due to random noise.

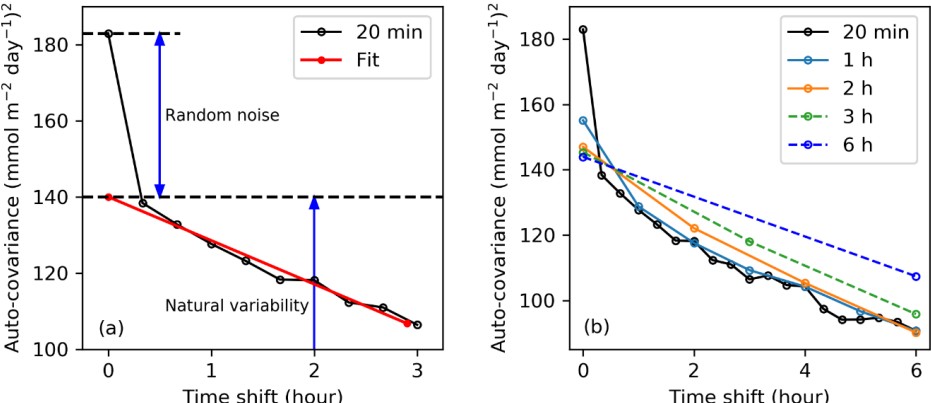

**Figure 7.** (a) Auto-covariance of the original 20-min fluxes (cruise JR18007) and a fit to the noise-free auto-covariance function extrapolated back to a zero time shift. (b) $CO_2$ flux auto-covariance functions with different averaging time scales. The black line represents the auto-covariance of the original 20-min fluxes. The 20-min fluxes are further averaged at different time scales (1, 2, 3 and 6 hour) and the corresponding auto-covariance functions are shown with different colours (dark blue, orange, green and light blue).

The random noise in the $CO_2$ fluxes decreases with a longer averaging time scale, with the greatest effect observed from 20 min to 1 hour (Fig. 7b). A fit to the noise-free auto-covariance function extrapolated back to a zero time shift gives us an estimate of the non-noise variability in the natural $CO_2$ flux. Subtracting the extrapolated natural flux variability from the total variance in $CO_2$ flux provides an estimate of the random noise in the flux for each averaging timescale (Fig. 7a). All four cruises consistently demonstrate a non-linear reduction in the noise contribution to the flux measurements when the averaging timescale increases (Fig. 8). The random noise in flux can be expressed relative to the natural variance in flux representing the inverse of the signal-to-noise ratio (i.e. random noise in flux/natural flux variability , hereafter referred to as noise: signal).





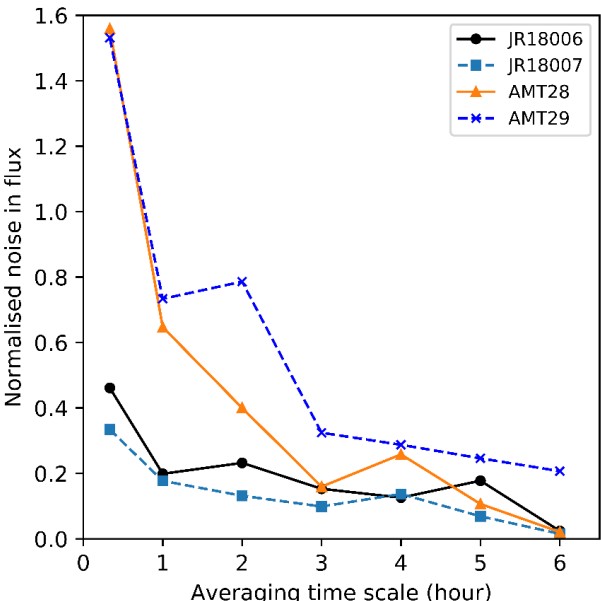


**Figure 8.** Effect of the averaging timescale on the noise: signal ( random noise in flux/ natural flux variability) for EC air-sea $CO_2$ flux measurements during four cruises.


The noise: signal also facilitates comparison of all four cruises (Fig. 8) and demonstrates the
consistent effect that increasing the averaging timescale has on noise: signal. Consistent with
Table 4, the Arctic cruises show much lower noise: signal because the flux magnitudes are
much  larger. Typical detection limits in analytical science are often defined by a 1: 3 noise:
signal ratio. A 1: 3 noise: signal is achieved with a 1 h averaging timescale for the Arctic cruises.
The Atlantic cruises encountered much lower air-sea $CO_2$ fluxes and an averaging timescale of
at least 3 h is required to achieve the same 1: 3 noise: signal ratio.
The flux measurement uncertainty at a 6-h averaging timescale for the AMT cruises is ~0.6
mmol m$^{-2}$ day$^{-1}$. The analysis presented above permits an answer to the question posed at the
beginning of the Results section. The mean difference between the 6-h averaged EC $CO_2$ flux
observations on AMT29 and AMT28 (1.3 mmol m$^{-2}$ day$^{-1}$, Fig. 4a) is much greater than the
measurement uncertainty. This significant difference was likely because of the interannual
variability in AMT $CO_2$ flux due to changes in the natural environment (e.g. $\Delta f CO_2$, sea surface
temperature, and physical drivers of interfacial turbulence such as wind speed) during the two
cruises.





At a typical research ship speed of ~10 knots, the AMT cruises cover ~110 km in 6 h, which is
equivalent to ~1° latitude. Averaging for longer than 6 h is likely to cause substantial loss of
real information about the natural variations in air-sea $CO_2$ flux and the drivers of flux
variability. For example, the mean flux between 0–20° S during cruise AMT28 is 0.9 mmol m$^{-2}$ day$^{-1}$. However, the 6 h average EC measurements show that the flux varied between +5 mmol
m$^{-2}$ day$^{-1}$ (~2–6° S) and -5 mmol m$^{-2}$ day$^{-1}$ (~11–13° S, Fig. 4a).

**4.2 Effect of $CO_2$ flux uncertainty on the gas transfer velocity $K$**

The uncertainties in the EC $CO_2$ air-sea flux measurement will influence the uncertainty that
translates to EC-based estimates of the gas transfer velocity, $K$. For illustration, $K$ is computed
for Arctic cruise JR18007, which had a high flux signal: noise ratio of ~5 (Fig. 8). Any data
potentially influenced by ice and sea ice melt were excluded using a sea surface salinity filter
(data excluded when salinity < 32). Equation 1 is rearranged and used with concurrent
measurements of $CO_2$ flux ($F$), $\Delta f CO_2$, and sea surface temperature (SST) to obtain $K$ adjusted
for the effect of temperature ($K_{660}$).
The determination coefficient ($R^2$) of the quadratic fit between wind speed ($U_{10N}$) and EC-
derived $K_{660}$ (Fig. 9) demonstrates that wind speed explains 76% of the $K_{660}$ variance during
Arctic cruise JR18007. How much of the remaining 24% can be attributed to uncertainties in
EC $CO_2$ fluxes?

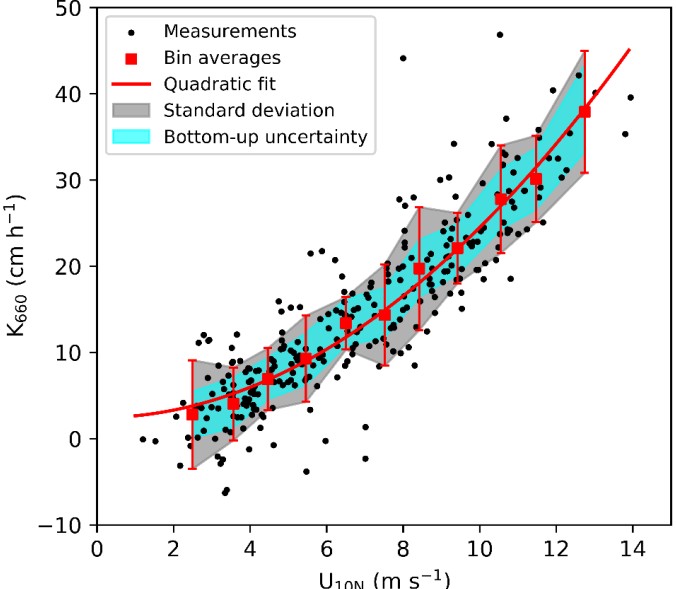


**Figure 9.** Gas transfer velocity ($K_{660}$) measured on Arctic cruise JR18007 (hourly average, signal: noise

~5) versus 10-m neutral wind speed ($U_{10N}$). Red squares represent 1 m s$^{-1}$ bin averages with error bars

representing one standard deviation (SD). The red curve represents a quadratic fit using the bin averages:

$K_{660} = 0.22U_{10N}^2 + 2.46$ (R$^2$ = 0.76). The grey shaded area represents the standard deviation calculated

for each wind speed bin ($K_{660} \pm 1SD$). The cyan region represents the upper and lower bounds in $K_{660}$

uncertainty computed from the EC flux uncertainty ($K_{660} \pm \delta K_{660}$, see text for detail).

552

Variability in $K_{660}$ within each 1 m s$^{-1}$ wind speed bin can be considered to have minimal wind

speed influence. It is thus useful to compare the variability within each wind speed bin ($K_{660} \pm$

1SD) with the upper and lower uncertainty bounds derived from the EC flux measurements.

Uncertainty in EC flux-derived $K_{660}$ ($\delta K_{660}$) is calculated from the uncertainty in hourly EC

flux ($\delta F$) by rearranging Eq. 1 (bulk flux equation) and replacing $F$ with $\delta F$. The resultant $\delta K_{660}$

is then averaged in wind speed bins. The shaded cyan band in Fig. 9 ($K_{660} \pm \delta K_{660}$) is

consistently narrower than the grey shaded band ($K_{660} \pm 1SD$). On average, EC flux-derived

uncertainty in $K_{660}$ can only account for a quarter of the $K_{660}$ variance within each wind speed

bin and the remaining variance is most likely due to the non-wind speed factors that influence

gas exchange (e.g. breaking waves, surfactants).



The analysis above can be extended to assess how EC flux-derived uncertainty affects our
ability to parameterise $K_{660}$ (e.g. as function of wind speed). To do so, a set of synthetic $K_{660}$
data is generated (same $U_{10N}$ as the $K_{660}$ measurements in Fig. 9). The synthetic $K_{660}$ data are
initialised using a quadratic wind speed dependence that matches JR18007 (i.e. $K_{660}$ =
$0.22U_{10N}^2 + 2.46$). Random Gaussian noise is then added to the synthetic $K_{660}$ data, with relative
noise level corresponding to the relative flux uncertainty values taken from JR18007 (mean of
20%, Table 4). The relative uncertainty in $K_{660}$ due to EC flux uncertainty ($\delta K_{660}/K_{660}$) shows
a wind speed dependence (Fig. S4a, Supplement), and the artificially-generated Gaussian noise
incorporates this wind speed dependence (Fig. S4b, Supplement). The $R^2$ of the quadratic fit to
the synthetic data as a function of $U_{10N}$ is 0.90 (the rest of the variance is due to uncertainty in
$K_{660}$). Since wind speed explains 76% of variance in the observed $K_{660}$, it can be inferred that
non-wind speed factors can account for 14% (i.e. (100-76)% - (100-90)%) of the total variance
in $K_{660}$ from this Arctic cruise. If the synthetic $K_{660}$ data is assigned a relative flux uncertainty
of 50% (reflective of a region with low fluxes, e.g. AMT28/29), the $R^2$ of the wind speed
dependence in the synthetic data decreases to 0.60.
The relative uncertainty in EC flux-derived $K_{660}$ ($\delta K_{660}/K_{660}$) is large when $|\Delta fCO_2|$ is small
(Fig. 10). Previous EC studies have filtered EC flux data to remove fluxes when the $|\Delta fCO_2|$
falls below a specified threshold (e.g. 20 µatm, Blomquist et al. (2017); 40 µatm, Miller et al.
(2010), Landwehr et al. (2014), Butterworth and Miller (2016), Prytherch et al. (2017); 50 µatm,
Landwehr et al. (2018)). Analysis of the data presented here suggests that a $|\Delta fCO_2|$ threshold
of at least 20 µatm is reasonable for hourly $K_{660}$ measurements, leading to $\delta K_{660}$ of ~10 cm h$^{-1}$
($\delta K_{660}/K_{660}$ ~1/3) or less on average. At very large $|\Delta fCO_2|$ of over 100 µatm, $\delta K_{660}$ is reduced
to only a few cm h$^{-1}$ ($\delta K_{660}/K_{660}$ ~1/5). At longer flux averaging time scales, it may be possible
to relax the minimal $|\Delta fCO_2|$ threshold.

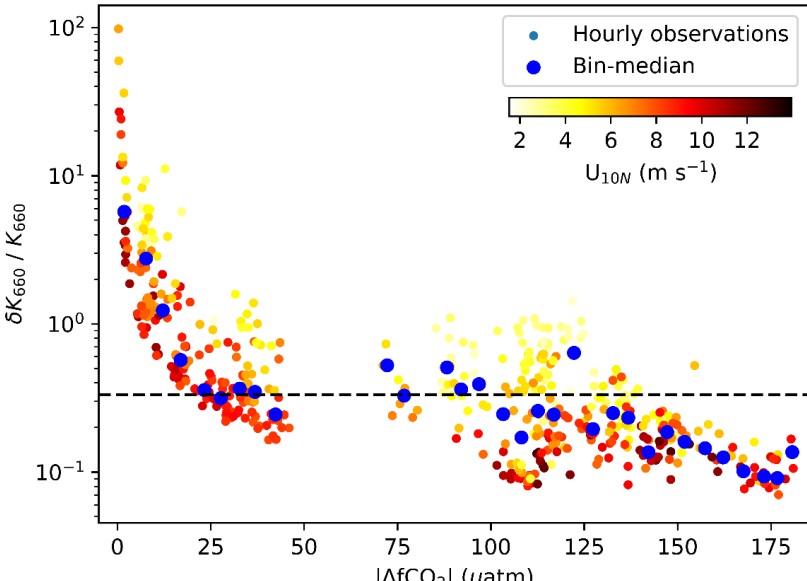


**Figure 10.** Relative uncertainty in EC-estimated hourly $K_{660}$ ($\delta K_{660}/K_{660}$) versus the magnitude of the air-sea $CO_2$ fugacity difference ($|\Delta f CO_2|$) during Arctic cruise JR18007 and Atlantic cruises AMT28 and AMT29 (no $\Delta f CO_2$ data were collected on JR18006). The data points are colour-coded by wind speed. Blue points are medians of $\delta K_{660}/K_{660}$ in 5 μatm bins. Here we use the parameterised $K_{660}$ (= $0.22 U_{10N}^2 + 2.46$) to normalise the uncertainty in $K_{660}$. The dashed line represents the 3: 1 signal: noise ratio ($\delta K_{660}/K_{660} = 1/3$).

594

## 5. Conclusions

This study uses data from four cruises with a range in air-sea $CO_2$ flux magnitude to comprehensively assess the sources of uncertainty in EC air-sea $CO_2$ flux measurements. Data from two ships and two different state-of-the-art $CO_2$ analysers (Picarro G2311-f and LI-7200, both fitted with a dryer) are analysed using multiple methods (Sect. 2.3). Random error accounts for the majority of the flux uncertainty, while the systematic error (bias) is small (Table 4). Random flux uncertainty is primarily caused by variance in $CO_2$ mixing ratio due to atmospheric processes. The random error due to instrument noise for the Picarro G2311-f is threefold smaller than for LI-7200 (Table 4 and Fig. D1, Appendix D). However, the contribution of the instrument noise to the total random uncertainty is much smaller than the



contribution of atmospheric processes such that both gas analysers are well suited for air-sea
$CO_2$ flux measurements.
The mean uncertainty in hourly EC flux is estimated to be 1.4–3.2 mmol m$^{-2}$ day$^{-1}$, which
equates to the relative uncertainty of ~20% in high $CO_2$ flux regions and ~50% in low $CO_2$ flux
regions. Lengthening the averaging timescale can improve the signal: noise ratio in EC $CO_2$
flux through the reduction of random uncertainty. Auto-covariance analysis of $CO_2$ flux is used
to quantify the optimal averaging timescale (Fig. 7 and 8, Sect. 4.1). The optimal averaging
timescale varies between 1 hour for regions of large $CO_2$ flux (Arctic in our analysis) and at
least 3 hours for regions of low $CO_2$ flux (tropical/sub-tropical Atlantic in our analysis).
The measurement uncertainty in EC $CO_2$ flux contributes directly to scatter in the derived gas
transfer velocity, $K_{660}$. Flux uncertainties determined in this paper are applied to a synthetic
$K_{660}$ dataset. This enables a partitioning of the variance in measured $K_{660}$ that is due to EC $CO_2$
flux uncertainty, wind speed, and other processes (10%, 76%, 14% for Arctic cruise JR18007).
At a given averaging timescale, a $|\Delta fCO_2|$ threshold helps to reduce the scatter in $K_{660}$. A
minimum $|\Delta fCO_2|$ filter of 20 μatm is needed for interpreting hourly $K_{660}$ data, with the signal:
noise ratio in $K_{660}$ improving further at higher $|\Delta fCO_2|$.


**Appendix A: Cruise tracks**



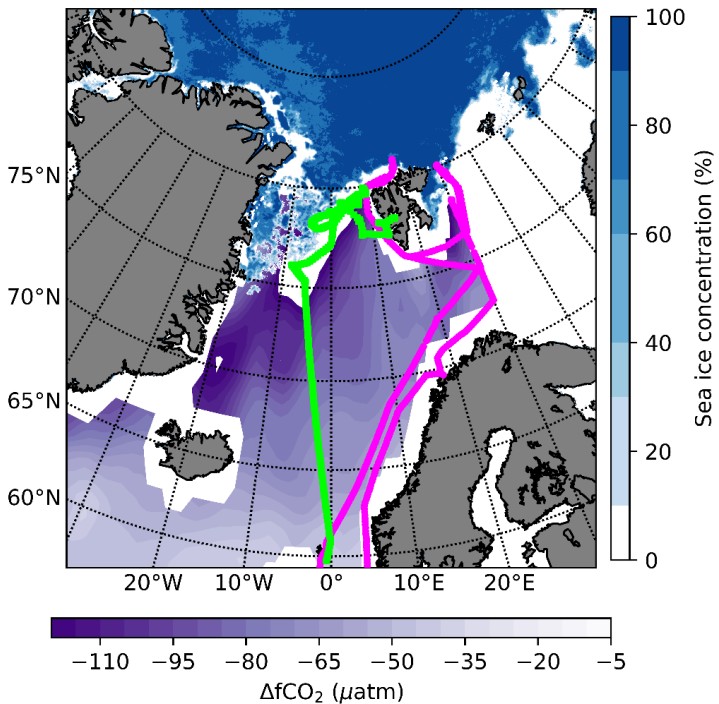

**Figure A1.** Cruise tracks of JR18006 (magenta) and JR18007 (green). The bottom colour bar indicates the $CO_2$ fugacity difference ($\Delta f CO_2$) of August 2019 (Bakker et al., 2016; Landschützer et al., 2020), while the right colour bar shows the Arctic sea ice concentrations of 1[st] August 2019 measured by Advanced Microwave Scanning Radiometer - Earth Observing System Sensor (AMSR-E, Spreen et al., 2008).





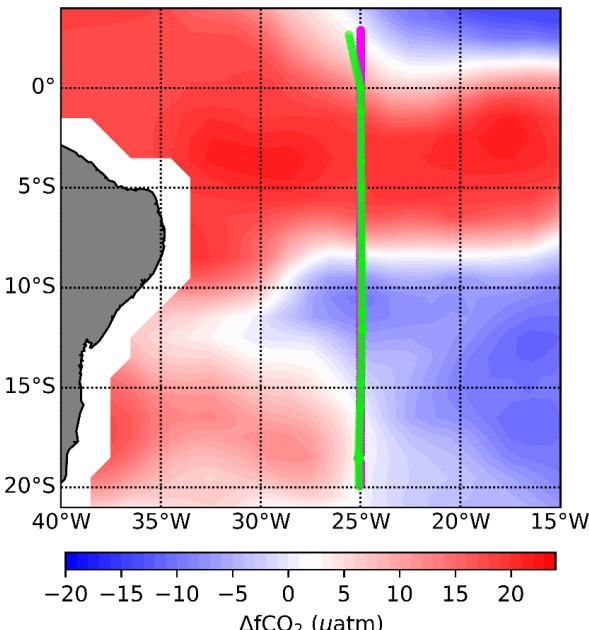

**Figure A2.** Cruise tracks of AMT28 (magenta) and AMT29 (green). The ocean is coloured with the
$\Delta f CO_2$ for October 2018 (Bakker et al., 2016; Landschützer et al., 2020).

**Appendix B: Integral time scale and variance spectra of $CO_2$ and vertical wind velocity**

Integral time scale is used in the flux uncertainty calculation (Eq. 5 and 7). The definition of
integral time scale $\tau_x$ of variable $x$ is:

$$\tau_x = \frac{1}{\sigma_x^2} \int_0^\infty r_{xx}(t) dt \qquad (B1)$$

where $\sigma_x^2$ is the variance of $x$ and $r_{xx}$ is the auto-covariance function of $x$. $t$ is the shifting time
of auto-covariance (which is different from the lag time between $w$ and $CO_2$ in the EC flux
calculation). We can use Eq. B1 to estimate the integral time scale of $w$ and $CO_2$ directly.
However, integration up to infinity is not practical. Instead we can numerically estimate the
time scale by determining the time corresponding to the auto-covariance coefficient function
$(r_{xx}/\sigma_x^2)$ value decaying to 1/e (1/e decaying method) or by integrating the auto-covariance
function up to the first zero crossing of the function (zero crossing method) (Rannik et al.,
2009).





One can also use similarity theory to estimate the integral time scale theoretically (Blomquist
et al., 2010):

$$\tau_w = 2.8 \frac{z}{\overline{u_r}} f_\tau(z/L) \tag{B2}$$

Here, $\overline{u_r}$ is the relative wind speed. The similarity function $f_\tau(z/L)$ is described by the stability
parameter $z/L$ where $z$ is the observation height (m) and $L$ is the Obukhov length (m)
(Blomquist et al., 2010).
Yet another method to estimate the integral time scale is from the peak frequency ($f_{max}$) in the
$w$ variance spectrum (Kaimal and Finnigan, 1994):

$$\tau_w = \frac{1}{2\pi f_{max}} \tag{B3}$$

The integral time scales of $w$ estimated by these four methods for cruise JR18007 are shown in
Figure B1. The integral time scale estimated by the zero crossing method agrees well with the
peak frequency estimates using Eq. B3. The 1/e decaying method tends to underestimate the
integral time scale, which is generally observed for turbulent signals (Rannik et al., 2009),
whereas the similarity method (Eq. B2) considerably overestimates the integral time scale. In
this study we use the integral time scale of $w$ from the zero crossing method to estimate the
theoretical flux uncertainty (Eq. 5 and 7). The theoretical systematic error estimates (Eq. 8)
also require the integral time scale of $CO_2$. The integral time scale of $CO_2$ is difficult to evaluate
from the above four methods due to instrument noise. Instead, we estimate it by directly
integrating the auto-covariance function (Eq. B1) to a shift time of 200 s (we found no
significant difference of the integral time scale when integrating the $CO_2$ auto-covariance
function for shift times ranging from 150 s to 250 s).





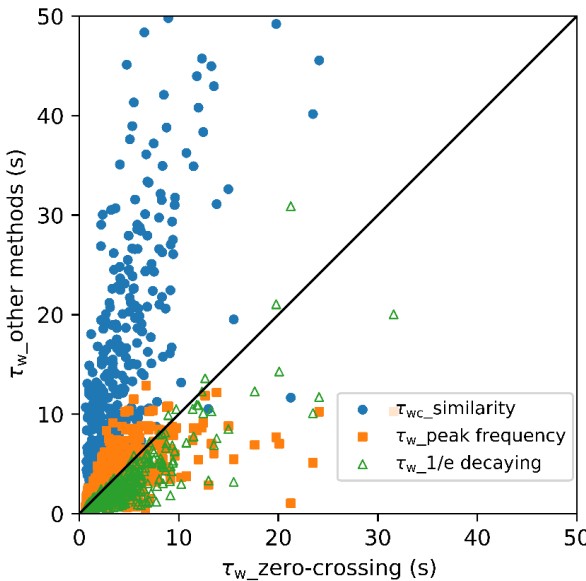


**Figure B1.** Comparison of integral time scales of *w* estimated by four different methods. Estimated
integral time scales from the zero crossing method (integrating the auto-covariance function up to first
zero crossing the function) agree well with the estimation of peak frequency method (Eq. B2). However,
the similarity method (Eq. B1) overestimates the integral time scale whereas the 1/e decaying method
(determining the time needed for the auto-covariance coefficient function value to decay to 1/e) tends
to underestimate the integral time scale.





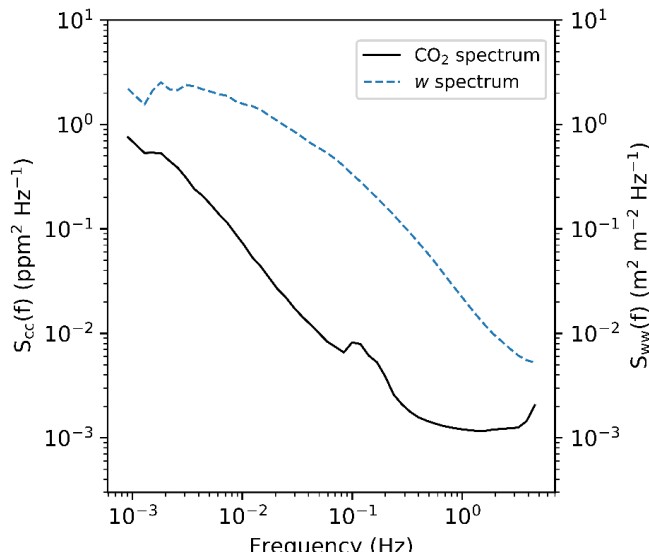


**Figure B2.** Mean variance spectra for $CO_2$ and $w$ for one Arctic cruise JR18007. The near constant $CO_2$

variance at high frequency (1-5 Hz) indicates the band-limited noise in the $CO_2$ signal. In contrast, the

$w$ spectrum does not show a similar band-limited noise at < 10 Hz.


**Appendix C: Comparison of the uncertainty estimates by different methods**




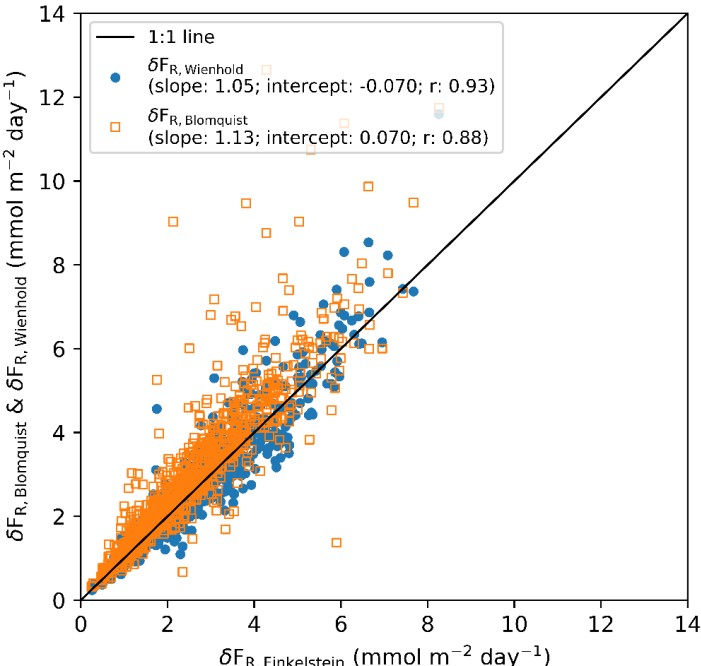


**Figure C1.** Comparison of total random uncertainties in hourly flux estimated by three different methods for the Arctic cruises. The empirical estimates $F_{R,\text{Wienhold}}$ agree well with one of the theoretical estimates $\Delta F_{R,\text{Finkelstein}}$ (r = 0.93). The other theoretical estimate $\Delta F_{R,\text{Blomquist}}$ is slightly higher than the random uncertainties $\Delta F_{R,\text{Finkelstein}}$ (slope = 1.13) if the constant in Eq. 8 is set equal to $\sqrt{2}$.

688



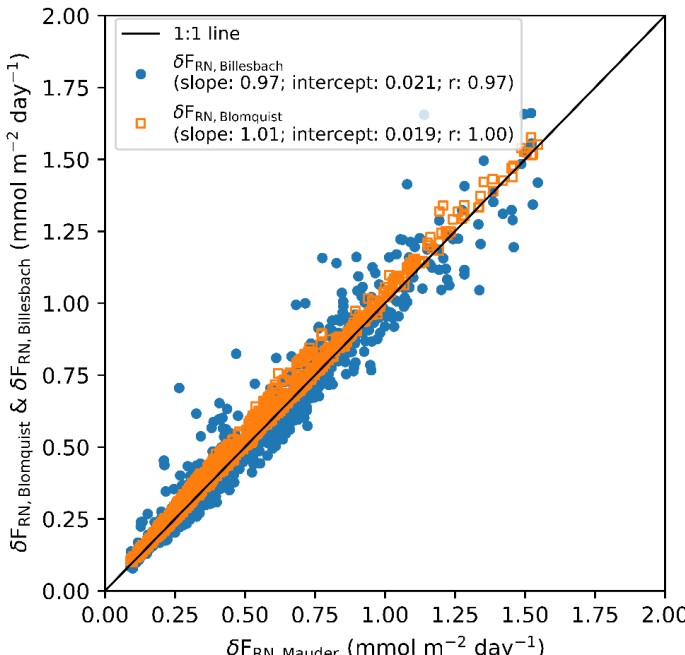

689

**Figure C2.** Comparison of random error in hourly flux due to instrument white noise, estimated by three different methods for the Arctic cruises. The three uncertainty estimations agree well. The correlation coefficient (r) between $\delta F_{RN, Mauder}$ and $\delta F_{RN, Blomquist}$ is 1 if the constant in Eq. 7 ($a$) is set to $\sqrt{2}$.

694

**Appendix D: Performance of two gas analysers**

Figure D1 shows a comparison between the performance of the Picarro 2311-f and the LI-7200 gas analysers. We estimated that the noise of the LI-7200 is on average 3 times higher than that of the Picarro 2311-f (Table 3). Indeed, random error in the $CO_2$ flux due to the white noise is much higher for the LI-7200 than for the Picarro 2311-f, but the total flux uncertainty of the EC system with the LI-7200 on AMT29 is only slightly higher than that of the EC system with the Picarro 2311-f on AMT28 (Table 4). Again, this is because for both EC systems, sampling error dominates the total random uncertainty, while the contribution of instrument noise (< 30%) to the total uncertainty is relatively small (Billesbach, 2011; Langford et al., 2015; Mauder et al., 2013; Rannik et al., 2016). Another often used CRDS gas analyser in EC measurements is the Los Gatos Research (LGR) Fast Greenhouse Gas Analyser (FGGA)





(Prytherch et al., 2017). Yang et al. (2016) showed that LGR FGGA is ca. 10 times noisier than
the Picarro G2311-f, and as a result the total $CO_2$ flux uncertainty measured by the LGR is 4
times higher than that by the Picarro. From the perspective of measurement noise, Picarro and
LI-7200 gas analysers are better suited for air-sea $CO_2$ flux measurements than the LGR FGGA.

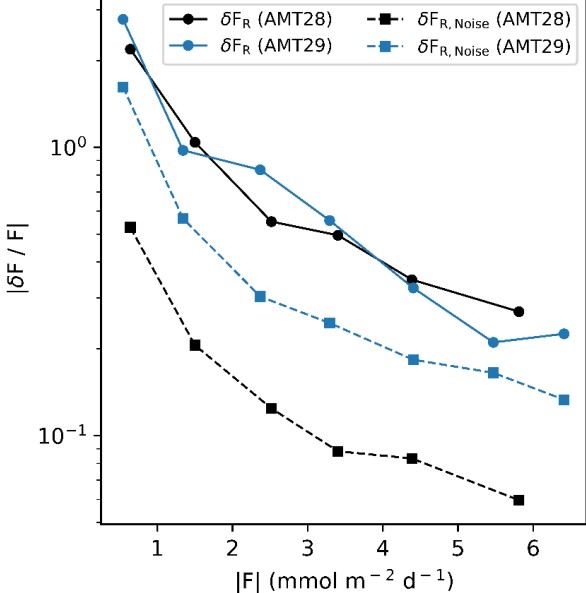


**Figure D1.** Comparison of the relative total random uncertainty and the relative random error
component due to white noise for different gas analysers. A Picarro G2311-f gas analyser was used on
AMT28 and a LI-7200 infrared gas analyser on AMT29.


*Data availability.* The processed hourly EC $CO_2$ fluxes and uncertainties can be found in the
Supplement of this paper. Raw, high frequency (10 Hz) data are large (tens of gigabytes) and are
archived at PML. Please contact the authors directly if you are interested in the raw data.

*Supplement.* The supplement related to this article is available online at:



*Author contributions.* TB and MY designed and installed the eddy covariance systems on ships and managed the collections of measurements. VK collected and processed the $CO_2$ fugacity data. YD processed and analysed the data with the help of MY and TB. YD wrote the paper with input from DB, TB and MY. All authors contributed to and approved the final manuscript.

*Competing interests.* The authors declare that they have no conflict of interest.

*Acknowledgements.* We thank captains and crew of the RRS James Clark Ross and RRS Discovery and all those who have helped keep the $CO_2$ flux systems running. We are extremely grateful to B. J. Butterworth (University of Calgary) for his advice on how to setup and run the automated $CO_2$ flux system on JCR and how to code the CR6 data logger, as well as to T. J. Smyth (PML) for setting up the remote monitoring of flux data. We also greatly appreciate I. Brown (PML) and D. Phillips for $fCO_2$ measurements and P. S. Liss (UEA) for support and helpful comments.

*Financial support.* This work is funded by the China Scholarship Council (CSC/201906330072). Air-sea $CO_2$ flux measurements were facilitated by European Space Agency (ESA AMT4oceanSatFlux project, grant no. 4000125730/18/NL/FF/gp) and support from the Natural Environment Research Council (NERC) via PML's contribution to the ORCHESTRA program (NE/N018095/1). The Arctic cruises were also supported by NERC, through the DIAPOD (NE/P006280/2) and ChAOS (NE/P006493/1) projects.

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
