# Peer review of "Uncertainties in eddy covariance air-sea CO2 flux measurements and"

_Atmospheric Chemistry and Physics, 2021_

## Referee Comment (RC1)

This manuscript presents an analysis of random error in the measurement of eddy correlation trace gas fluxes at sea and the effects of measurement error on the interpretation of direct air-sea gas transfer velocity observations.  The analysis employs field measurements from four research cruises – two N-S transects of the Atlantic and two high-latitude Arctic projects.  The study includes two state-of-art measurement systems for carbon dioxide flux – a broadband infrared gas analyzer (IRGA, LI-7200) and a laser cavity ring-down spectrometer (CRDS, Picarro G2311-f).

This manuscript provides a very useful overview of various approaches developed over the years to assess random error in flux measurements and analyzes these methods under conditions where the covariance signal is often near the measurement detection limit in the presence of various interferences such as platform motion, flow distortion and large water vapor fluxes.

The paper is very well written and well organized.  I don't have significant comments with respect to usage or punctuation and will confine the following comments to a few issues of substance.  Overall, this is a very good paper and a welcome contribution to the field and should be accepted after checking a few of the issues mentioned below.

I am not sure about the merits of the CO2/H2O decorrelation of the LI-7200 data, described on lines 183-185 (based on Landwehr et al. 2018).  This procedure has potential to remove real turbulent flux signal for CO2 since the water vapor and CO2 fluxes are both driven by the same turbulent eddies, and therefore correlated with each other.  L2018 state that due to the long inlet lag time and air drier in their configuration the gas signals are decoupled from the vertical wind measurements (which is true) and therefore this decorrelation doesn't remove real flux signal (which I'm not sure about).  The decorrelation applied here is not with respect to vertical wind – it is between the two gas concentrations measured simultaneously by the same analyzer.  If these signals have approximately the same lag time, then it seems to me the decorrelation could indeed remove actual CO2 flux signal by removing variance due to low-frequency turbulent eddies present in both signals which pass through the air drier (the drier is basically a low-pass filter on the water vapor signal).  Did this decorrelation yield a significant adjustment to the measured fluxes?  If not, maybe it's unnecessary.

Did the authors check for a positive bias to the CO2 fluxes due to the demonstrated crosstalk between water vapor and CO2 signals in the IRGA?  The use of a drier to precondition the sample air is necessary to remove this artifact, and I'm sure the authors approach is fairly effective in this respect.  But it might be useful to check the correlation/covariance/cospectra of the water vapor and vertical wind signals on AMT29 to see if low-frequency latent heat flux signal is nevertheless bleeding through the drier and affecting the CO2 measurement (as mentioned above).  (Note, the lag time adjustment may be a bit different for water vapor and CO2.)  I mention this because the AMT cruises are the primary comparison between the two methods and the corrected IRGA CO2 fluxes on AMT29 are a bit larger than those from the CRDS on AMT28, which is what you might expect if water vapor cross talk is bleeding into the IRGA CO2 flux measurement (at equatorial latitudes where we expect large latent heat fluxes!).

Of course, there could be other reasons for the observed difference between cruises separated by a year, as mentioned by the authors on p.23. It's a shame both analyzers were not deployed simultaneously on one of the cruises.

There seems to be an error in equation 5. Flux uncertainty goes as the square root of sampling time and the entire fractional term on the RHS of this equation should be to the ½ power. The authors have chosen to use the square root of the product of the two integral time scales in the numerator, which is different from the more common minimum value of the two integral scales, but this is OK. The missing square root may be just a typo, but if this equation was in fact used to estimate error, then that should be recomputed.

I can provide an update for the discussion of the integral time constant in Appendix B. Equation B2 in this manuscript and the associated stability function (both from Blomquist et al. 2010) are a bit dated. They were based on measurements from R/P Flip during the SCOPE field campaign and do not include much information for stable conditions. A more recent analysis (as yet unpublished) of the entire NOAA PSL flux database (41 research cruises spanning 21 years) has updated the empirical relationship for $\tau$ as a function of the nondimensional frequency maximum of the cospectrum, $\eta_m$

$$\tau = \frac{z}{2\pi U_r \eta_m}$$

Where the best fit for $\eta_m$ as a function of $z/L$ is given by

$$\eta_m(z/L) = A1 + \frac{1}{A2 + A3|z/L|} \quad for \ z/L < 0$$

$$\eta_m(z/L) = B1 + B2(z/L)^{2/3} \quad for \ z/L > 0$$

And the best-fit constants $A$ and $B$ differ for momentum and scalar fluxes:

|  | A1 | A2 | A3 | B1 | B2 |
|---|---|---|---|---|---|
| $\eta_m \ (w'u')$ | 0.033 | 25 | 400 | 0.069 | 0.42 |
| $\eta_m \ (w't')$ | 0.06 | 13 | 120 | 0.134 | 0.16 |
| $\eta_m \ (w'q')$ | 0.06 | 33 | 120 | 0.089 | 0.20 |

I've attached a plot below, where the black dashed line represents Equation B2 and the green line is the updated scalar flux integral time constant from the equations above. $U_r$ differs a bit between $z/L$ bins in the flux database, which causes a little scatter in the trend of each line, but it's clear the updated function in green yields a time constant considerably smaller than Equation B2 in black, especially in stable conditions, and this is more or less in agreement with what is shown in Figure B1 of this manuscript.

[Figure]

I'm not suggesting you include all this in the manuscript, but you can mention that based on recent analysis the equation B2 formulation is now thought to be an overestimate.

Note, your figure caption for Fig B1 has a couple typos: the peak frequency equation is B3 and the similarity relationship is B2.

I'm convinced the authors have demonstrated their principal conclusion – that for state-of-art gas analyzers sampling error is a more important contributor to flux uncertainty than analyzer noise, and this is the reason why we usually need to average over hourly timescales to achieve reasonable measurement precision. This is also why it is very difficult to make credible $CO_2$ flux measurements in the presence of significant turbulent disruptions and pollution plumes or other sources of $CO_2$ variability related to airmass advection. Threshold criteria for stationarity and homogeneity are sometimes also helpful in reducing measurement uncertainty.

---

## Author Comment (AC1)

*We thank the reviewer (Byron Blomquist) for his helpful comments and suggestions that helped us improve this manuscript. Below the reviewer comments are given in black. Our responses are given in red, and the updated text is given in blue.*

This manuscript presents an analysis of random error in the measurement of eddy correlation trace gas fluxes at sea and the effects of measurement error on the interpretation of direct air-sea gas transfer velocity observations. The analysis employs field measurements from four research cruises – two N-S transects of the Atlantic and two high-latitude Arctic projects. The study includes two state-of-art measurement systems for carbon dioxide flux – a broadband infrared gas analyzer (IRGA, LI-7200) and a laser cavity ring-down spectrometer (CRDS, Picarro G2311-f).

This manuscript provides a very useful overview of various approaches developed over the years to assess random error in flux measurements and analyzes these methods under conditions where the covariance signal is often near the measurement detection limit in the presence of various interferences such as platform motion, flow distortion and large water vapor fluxes.

The paper is very well written and well organized. I don't have significant comments with respect to usage or punctuation and will confine the following comments to a few issues of substance. Overall, this is a very good paper and a welcome contribution to the field and should be accepted after checking a few of the issues mentioned below.

I'm convinced the authors have demonstrated their principal conclusion – that for state-of-art gas analyzers sampling error is a more important contributor to flux uncertainty than analyzer noise, and this is the reason why we usually need to average over hourly timescales to achieve reasonable measurement precision. This is also why it is very difficult to make credible $CO_2$ flux measurements in the presence of significant turbulent disruptions and pollution plumes or other sources of $CO_2$ variability related to airmass advection. Threshold criteria for stationarity and homogeneity are sometimes also helpful in reducing measurement uncertainty.

Major Comments 1:

I am not sure about the merits of the $CO_2/H_2O$ decorrelation of the LI-7200 data, described on lines 183-185 (based on Landwehr et al. 2018). This procedure has potential to remove real turbulent flux signal for $CO_2$ since the water vapor and $CO_2$ fluxes are both driven by the same turbulent eddies, and therefore correlated with each other. L2018 state that due to the long inlet lag time and air drier in their configuration the gas signals are decoupled from the vertical wind measurements (which is true) and therefore this decorrelation doesn't remove real flux signal (which I'm not sure about). The decorrelation applied here is not with respect to vertical wind – it is between the two gas concentrations measured simultaneously by the same analyzer. If these signals have approximately the same lag time, then it seems to me the decorrelation could indeed remove actual $CO_2$ flux signal by removing variance due to low frequency turbulent eddies present in both signals which pass through the air drier (the drier is basically a low-pass filter on the water vapor signal). Did this decorrelation yield a significant adjustment to the measured fluxes? If not, maybe it's unnecessary.

Answer: The decorrelation is between the concurrent $CO_2$ signal and $H_2O$ signal (i.e. $CO_2$ signal and the $H_2O$ signal sensed by the LI-7200 at the same time). Even without a Nafion dryer, the lag time for $H_2O$ should be much longer than the lag time for $CO_2$ (Figure 9 in Yang et al., 2016) because the polar $H_2O$ molecular is much 'stickier' than $CO_2$ and tends to adsorb onto the wall of the tubing. Therefore, for our setup with a dryer, we do not expect the $CO_2$:$H_2O$ decorrelation to remove much real turbulent flux signal in $CO_2$ because 1) there shouldn't be much H2O flux remaining (Figure R1), and 2) the $CO_2$ signal is decoupled from the $H_2O$ signal.

Figure R1 and R2 are examples from the AMT29 cruise in the tropics (LI-7200 setup had a shorter inlet tube and the data is thus more likely to be impacted). Figure R1 shows that the $H_2O$ variance is small at the high frequency domain, but the variance is quite large at the low frequency domain ($< 5 \times 10^{-3}$ Hz). However, seems this low frequency variance in $H_2O$ is not the real flux signal because the behaviour of the low frequency $H_2O$:W cospectrum (computed at the lag of $CO_2$) is similar to the cospectrum at the high frequency. To clarify, the peak value of the $H_2O$:W cospectrum at ~0.1 Hz is due to the ship motion.

Figure R2 shows that the $CO_2$:W cospectrum is only very slightly different with and without the $CO_2$:$H_2O$ decorrelation. We think this difference is due to the influence of variability in $H_2O$ that is not vertical flux. Therefore, we think the $CO_2$:$H_2O$ decorrelation used in the manuscript is acceptable.

[Figure]

Figure R1. The mean 20 min cospectrum of $H_2O$:W ($H_2O$ at the $CO_2$ lag time) and $H_2O$ variance spectrum on 5 November 2019 (time, 18:00–23:00; latitude, 2.54°S–3.20°S; mean wind speed, 8.00 ± 0.42 m s$^{-1}$).

[Figure]

Figure R2. The mean 20 min $CO_2$:W cospectra before and after the $CO_2$:$H_2O$ decorrelation on 5 November 2019 (time, 18:00–23:00; latitude, 2.54°S–3.20°S; mean wind speed, 8.00 ± 0.42 m s-1).

The effect of $H_2O$ decorrelation on the LI-7200 $CO_2$ flux is fairly small. Table R1 shows the mean of the $CO_2$ flux magnitude and the variance of the $CO_2$ flux during the entire cruise of AMT29 (LI-7200 was used). The $CO_2$:$H_2O$ decorrelation slightly reduces the magnitude of the $CO_2$ flux (by an average of 7%) and the variance of the hourly flux (by an average of 14%). Figure R3 shows the comparison of the hourly $CO_2$ flux with and without the $CO_2$:$H_2O$ decorrelation.

Table R1. $CO_2$ flux during the entire cruise of AMT29.

| $CO_2$ flux | With $H_2O$ decorrelation | Without $H_2O$ decorrelation |
|---|---|---|
| \|Mean\| (mmol m$^{-2}$ s$^{-1}$) | 4.89 | 5.23 |
| Variance (mmol m$^{-2}$ s$^{-1}$)$^2$ | 48.14 | 54.78 |

[Figure]

Figure R3. Hourly $CO_2$ flux without the $CO_2$:$H_2O$ decorrelation versus the flux with the $CO_2$:$H_2O$ decorrelation during the entire cruise of AMT29.

Major Comments 2:

Did the authors check for a positive bias to the $CO_2$ fluxes due to the demonstrated crosstalk between water vapor and $CO_2$ signals in the IRGA? The use of a drier to precondition the sample air is necessary to remove this artifact, and I'm sure the authors approach is fairly effective in this respect. But it might be useful to check the correlation/covariance/cospectra of the water vapor and vertical wind signals on AMT29 to see if low-frequency latent heat flux signal is nevertheless bleeding through the drier and affecting the $CO_2$ measurement (as mentioned above). (Note, the lag time adjustment may be a bit different for water vapor and $CO_2$.) I mention this because the AMT cruises are the primary comparison between the two methods and the corrected IRGA $CO_2$ fluxes on AMT29 are a bit larger than those from the CRDS on AMT28, which is what you might expect if water vapor cross talk is bleeding into the IRGA $CO_2$ flux measurement (at equatorial latitudes where we expect large latent heat fluxes!).

Of course, there could be other reasons for the observed difference between cruises separated by a year, as mentioned by the authors on p.23. It's a shame both analyzers were not deployed simultaneously on one of the cruises.

Answer: As shown in Figure R1 and R2, we think there is no obvious residual low-frequency latent heat flux signal after the air sample was dried. However, the variability in $H_2O$ not due to vertical flux might still affect the $CO_2$ flux measurements. We addressed this issue by decorrelating the $CO_2$ signal against the $H_2O$ signal. The decorrelation reduces the $CO_2$ flux only slightly, and it cannot explain the larger $CO_2$ fluxes on AMT29.

As stated on p23 in the manuscript and shown in Figure R4, We think the difference in $CO_2$ flux between the two AMT cruises is mostly due to natural variability (AMT28: 9 October–16 October 2018; AMT29: 4 November–11 November 2019). Figure R4 shows that the main reason for the greater (more positive) $CO_2$ flux during AMT29 than AMT28 is likely due to the difference in d$f$CO$_2$.

[Figure]

Figure R4. The distributions of the $CO_2$ fugacity difference between the sea surface and the overlying atmosphere (d$f$CO$_2$) and wind speed (U10n) against the latitude during cruises AMT28 and AMT29.

Major Comments 3:

There seems to be an error in equation 5. Flux uncertainty goes as the square root of sampling time and the entire fractional term on the RHS of this equation should be to the ½ power. The authors have chosen to use the square root of the product of the two integral time scales in the numerator, which is different from the more common minimum value of the two integral scales, but this is OK. The missing square root may be just a typo, but if this equation was in fact used to estimate error, then that should be recomputed.

Answer: Yes, the random flux uncertainty goes as the square root of sampling time (equation 6 and 7 in the manuscript) and the minimum value of the two integral tine scales. However, for the bias (systematic error), it is different. If we look at the bias estimation equation 28 and the random error estimation equation 49 in Lenschow et al. (1994), you can see for bias (equation

28), the exponential of the sampling time $T$ is 1; but the random error goes as the square root of sampling time (equation 49). We think this difference is because of the different derivative processes. The random error is derived from the error variance of the flux, while the bias is derived from the direct difference between the ensemble averaged flux and the time averaged flux (see Lenschow et al. (1994) for the detailed derivative processes).

$$\frac{|F - \langle F(T) \rangle|}{(\mu_2 \mu_s)^{1/2}} \leqslant 2 \frac{(TT_s)^{1/2}}{T} . \qquad (28)$$

$$\frac{\sigma_F(T)}{|F|} \leqslant \frac{2}{r_{ws}} \left[ \frac{\min(T, T_s)}{T} \right]^{1/2}, \qquad (49)$$

Major Comments 4:

I can provide an update for the discussion of the integral time constant in Appendix B. Equation B2 in this manuscript and the associated stability function (both from Blomquist et al. 2010) are a bit dated. They were based on measurements from R/P Flip during the SCOPE field campaign and do not include much information for stable conditions. A more recent analysis (as yet unpublished) of the entire NOAA PSL flux database (41 research cruises spanning 21 years) has updated the empirical relationship for $\tau$ as a function of the nondimensional frequency maximum of the cospectrum, $\eta_m$

$$\tau = \frac{z}{2\pi U_r \eta_m}$$

Where the best fit for $\eta$! as a function of $z/L$ is given by

$$\eta_m(z/L) = A1 + \frac{1}{A2 + A3|z/L|} \quad for \ z/L < 0$$

$$\eta_m(z/L) = B1 + B2(z/L)^{2/3} \quad for \ z/L > 0$$

And the best-fit constants $A$ and $B$ differ for momentum and scalar fluxes:

|  | A1 | A2 | A3 | B1 | B2 |
|---|---|---|---|---|---|
| $\eta_m (w'u')$ | 0.033 | 25 | 400 | 0.069 | 0.42 |
| $\eta_m (w't')$ | 0.06 | 13 | 120 | 0.134 | 0.16 |
| $\eta_m (w'q')$ | 0.06 | 33 | 120 | 0.089 | 0.20 |

I've attached a plot below, where the black dashed line represents Equation B2 and the     green line is the updated scalar flux integral time constant from the equations above. $U_r$ differs a bit between $z/L$ bins in the flux database, which causes a little scatter in the trend of each line, but it's clear the updated function in green yields a time constant considerably smaller than Equation B2 in black, especially in stable conditions, and this is more or less in agreement with what is shown in Figure B1 of this manuscript.

[Figure]

I'm not suggesting you include all this in the manuscript, but you can mention that based on recent analysis the equation B2 formulation is now thought to be an overestimate.

Note, your figure caption for Fig B1 has a couple typos: the peak frequency equation is B3 and the similarity relationship is B2.

Answer: Thanks. This update is very helpful. We were also confused by equation 2. Based on our cruise data analysis, the integral time scales estimated by equation B2 are much higher than the estimates by equation B1 and B3 (Figure B1 in the manuscript). Since the updated integral time scale (blue line in the above figure) is more or less in agreement with what is shown in Figure B1 of our manuscript, we will use the integral time scale estimated by equation B1 as we have done in our manuscript for the uncertainty calculation. For equation B2, we add a sentence in Appendix B to show the update.

Based on the recent analysis (as yet unpublished) of the entire NOAA PSL flux database, the Eq. B2 formulation is now thought to be an overestimate (review comment for this paper from B. Blomquist, 2021).

**Reference**

Lenschow, D. H., Mann, J. and Kristensen, L.: How long is long enough when measuring fluxes and other turbulence statistics?, J. Atmos. Ocean. Technol., 11(3), 661–673, doi:10.1175/1520-0426(1994)011<0661:HLILEW>2.0.CO;2, 1994.

Yang, M., Prytherch, J., Kozlova, E., Yelland, M. J., Parenkat Mony, D. and Bell, T. G.: Comparison of two closed-path cavity-based spectrometers for measuring air-water $CO_2$ and $CH_4$ fluxes by eddy covariance, Atmos. Meas. Tech., 9(11), 5509–5522, doi:10.5194/amt-9-5509-2016, 2016.

---

## Author Comment (AC2)

*We thank the anonymous reviewer for helpful comments that helped us improve this manuscript. Below the reviewer comments are given in black. Our responses are given in red, and the updated text is given in blue.*

**General Comments:**

This paper presents a comprehensive analysis of uncertainties in eddy covariance flux measurements over the ocean, discussing in detail the separation of systematic from random noise with a thorough overview of different methods for estimating the latter. The authors clearly demonstrate that with current top-end commercial gas analyzers, instrumental noise no longer contributes significantly to the flux uncertainty, this being dominated by variance associated with the nature of turbulent transport. The paper is very well written and structured, and may serve as an excellent reference for researchers looking for a state-of-the-art starting point for uncertainty analysis.

**Specific Comments:**

Line 40, Eq 1: perhaps briefly explain the "660" for those readers unfamiliar with gas transfer velocity parameterizations

Answer: we added one sentence to explain 660 after the definition of *Sc* (Schmidt number).

*Sc* is equal to 660 for $CO_2$ at 20°C and 35‰ salt water (Wanninkhof et al., 2009).

L76: Li-COR Inc. USA

Answer: we added the company information of LI-7500.

After 2000, a commercial open-path infrared gas analyser LI-7500 (Li-COR Inc. USA) became widely used for air-sea $CO_2$ flux measurements…

L79: expected fluxes

Answer: we replaced 'expected' by 'expected fluxes'.

The LI-7500 generated extremely large and highly variable $CO_2$ fluxes in comparison to expected fluxes…

L80: … 2011). This problem is generally considered …

Answer: we replaced '…2011), which are generally considered…' by '…2011). This problem is generally considered…'.

… 2011). This problem is  generally considered to be an artefact caused by water vapour cross-sensitivity…

L124: Comment on the deflection of the streamlines from horizontal and effects on the vertical wind component

Answer: CFD modelling suggests that at the location of the sonic anemometer on the JCR, the angle of air flow relative to the horizontal is 3.5 deg for bow on wind. This is consistent with the tilt angle computed from the double rotation as a part of the motion correction.  The tilt angle is slightly larger on the Discovery (7.5 deg) for bow on wind. This deflection of the streamline is accounted for by the double rotation prior to the EC flux calculation. Therefore, we add a sentence in the end of this paragraph:

The deflection of the streamline from horizontal and effects on the vertical wind component is accounted for by the double rotation (motion correction processes, see Sect. 2.2.) prior to the EC flux calculation for both ships.

L133: add the Reynolds number for completeness

Answer: the Reynolds numbers are 5957 and 1042 for the air sample tube on JCR and Discovery, respectively. We added this numbers. In addition, the inner diameter of the tube on Discovery is 0.95 cm, we also added this value. We changed the previous sentence 'Air is pulled through a long tube (30 m, 0.95 cm inner diameter) with a dry vane pump at a flow rate of ~40 L min-1 (Gast 1023 133 series).' by:

Air is pulled through a long tube (30 m, 0.95 cm inner diameter, Reynolds number 5957) with a dry vane pump at a flow rate of ~40 L min-1 (Gast 1023 133 series).

We changed the previous sentence 'The LI-7200 gas analyser was mounted within the enclosed staircase, directly underneath the meteorological platform and close to the inlet (inlet length 7.5 m).' by:

The LI-7200 gas analyser was mounted within the enclosed staircase, directly underneath the meteorological platform and close to the inlet (inlet length 7.5 m, inner diameter 0.95 cm, Reynolds number 1042).

Fig. 3: adding a schematic explanation similar to that in Fig. 7a would clarify the method of separating the white noise from the total variance.

Answer: we replaced the previous Fig. 3 and caption by the figure and caption below:

[Figure]

**Figure 3.** Mean normalised auto-covariance functions of $CO_2$ and vertical wind velocity ($w$) by four different instruments. The magenta line represents a fit to the noise-free auto-covariance function of $CO_2$ (measured by Picarro) extrapolated back to a zero time shift. An example of the white noise and natural variability contributions to the total $CO_2$ (measured by Picarro) variance is indicated by two blue arrows. The sharp decrease of the $CO_2$ auto-covariance between the zero shift and the initial 0.1 s shift corresponds to the large contribution of white noise from the gas analysers. The LI-7200 is the noisier instrument. The noise contributions from the anemometers are relatively small ($< 10\%$).

L432: should be Eq. 8

Answer: $\delta F_{RV}$ is calculated by Eq. 7 by setting $\sigma^2_{c_n} = 0$ and $\sigma^2_{c_a} = \sigma^2_{c_{av}}$. Here $\sigma^2_{c_{av}}$ is estimated by Eq. 8. We added Eq. 8 as another referred equation for calculating $\delta F_{RV}$.

…, vertical flux ($\delta F_{RV}$, Eq. 7 and 8) and other atmospheric processes…

L504 etc.: eliminate the space between : and signal; otherwise the colon looks like punctuation.

Answer: we eliminated all the space between : and signal (noise: signal) and also eliminated all the space between : and noise (signal: noise) in the manuscript (13 cases).

Signal:noise, noise:signal

Fig. A1: I recommend using color schemes that are more distinguishable from each other

for sea ice and dFCO2 (eg. a grey scale for the sea ice coverage).

Answer: we replaced the previous Fig. A1 by the below one. We tried to use a grey scale for the sea ice coverage, but it is difficult to distinguish the sea ice concentration and the land. Therefore, here we used the orange scale for the sea ice coverage. Seems it is better than the previous one (blue scale for the sea ice coverage).

[Figure]

L942: Chapt. 2

Answer: we corrected the wrong word.

Wyngaard, J. C.: Turbulence in the Atmosphere Part 1, Chapt. 2, Getting to know turbulence, p27-54 Cambridge University Press., 2010.

**Reference**

Wanninkhof, R., Asher, W. E., Ho, D. T., Sweeney, C., & McGillis, W. R. (2009). Advances in Quantifying Air-Sea Gas Exchange and Environmental Forcing. *Annual Review of Marine Science*, *1*(1), 213–244. https://doi.org/10.1146/annurev.marine.010908.163742